



# A 12-Year Climate Record of Wintertime Wave-Affected Marginal Ice Zones in the Atlantic Arctic based on CryoSat-2

Weixin Zhu[1], Siqi Liu[1], Shiming Xu[1,2], and Lu Zhou[3]

[1]Department of Earth System Science, Tsinghua University, Qinghuayuan 1, Haidian District, Beijing, China
[2]University Corporation of Polar Research, Beijing, China
[3]Department of Earth Science, University of Gothenburg, Guldhedsgatan 5a, Gothenburg, Sweden

**Correspondence:** Shiming Xu (xusm@tsinghua.edu.cn)

**Abstract.** Wave-affected marginal ice zone (MIZ) is an integral part of the sea ice cover and key to the atmosphere-ice-ocean interaction in the polar region. While we mainly rely on in-situ campaigns for studying MIZs, great challenges still exist for the remote sensing of the MIZs by satellites. In this study we develop a novel retrieval algorithm for wave-affected MIZs based on the delay-Doppler radar altimeter onboard CryoSat-2 (CS2). CS2 waveform power and the waveform stack statistics are used to determine the part of the ice cover affected by waves. Based on the CS2 data since 2010, we generate a climate record of wave-affected MIZs in the Atlantic Arctic, spanning 12 winters between 2010 and 2022. As indicated by the MIZ record, no significant change of either the mean MIZ width or the extreme width is detected, although large temporal and spatial variability is present. In particular, extremely wide MIZs (over 300 $km$) are observed in the Barents Sea, while in other part of the Atlantic Arctic, MIZs are generally narrower. We also compare the CS2-based retrieval with those based on the laser altimeter of ICESat2 and the synthetic aperture radar images from Sentinel-1. Under spatial and temporal collocation, we attain good agreement among the MIZ retrievals based on the three different types of satellite payloads. Moreover, the traditional sea ice concentration based definition MIZ yields systematically narrower MIZs than CS2, and there is no statistically significant correlation between the two. Besides CS2, the proposed retrieval algorithm can be adapted for various historical and future radar altimetry campaigns. The synergy of multiple satellites can further improve the spatial and temporal representation of the altimeters' observation of the MIZs.

## 1 Introduction

Marginal ice zone (MIZ) is the region of the sea ice edge that is affected by the open ocean (Wadhams, 2013). Waves and swells develop over open ocean, and propagate into the ice edge, with the ensuing sea ice break-up and the modification the floe sizes (Asplin et al., 2012). Consequently, thermodynamic, dynamic and coupled processes are incurred, such as frazil ice growth and latent heat release in the MIZ (Doble et al., 2015; Alberello et al., 2022). Besides, waves entering the sea ice gradually attenuate through frictional processes between ice floes and between the sea ice and the ocean. With the ongoing polar climate



changes (Stroeve and Notz, 2018), MIZs play even more important roles by inducing positive feedback on the sea ice cover (Asplin et al., 2012). Furthermore, human activities in polar regions are also closely related to the presence of the sea ice and
MIZs, such as fishing and navigation in polar waters (Ingvaldsen et al., 2021).

Although MIZs are important for both scientific research and marine operations, the direct observation of wave-affected MIZ is still very limited. In-situ campaigns in MIZs, in spite of the great challenges, provide us with the direct evidence of the wave's propagation and attenuation in the sea ice cover. However, in order to observe the MIZs at large scale, we mainly need satellite remote sensing techniques. A commonly used definition of the MIZ is the area with the satellite-observed sea ice
concentration (SIC) between 15% and 80% (Strong and Rigor, 2013), with the threshold value of 80% representing the 'closed ice' by the WMO's nomenclature. However, SIC products are usually generated from Passive Microwave Imaging (PMI) satellite payloads, which have limited spatial resolutions and the respective uncertainties. More importantly, the SIC-based MIZ definition does not reflect the ocean's processes that govern the MIZ, such as the wave propagation and interaction with the sea ice. For example, waves are found to propagate hundreds of kilometers into the fully-packed ice cover (i.e., SIC
close to 100%) during various in-situ campaigns (Kohout et al., 2020; Alberello et al., 2022). In this regard, there are growing efforts in the community for better and more physical definitions of the MIZs (Kohout et al., 2014; Horvat et al., 2020).

In order to explicitly resolve waves in the MIZ, high-resolution satellite payloads are usually needed, such as various imaging payloads and the laser altimetry of ICESat-2 (IS2) (Markus et al., 2017; Horvat et al., 2020; Collard et al., 2022). The effective footprint should be at least finer than half of the wavelength, which is no more than a few hundred meters. Besides, the
instantaneous observation of MIZs by satellites is further limited in terms of the temporal representation of the MIZs, mainly due to their highly variant nature. In general, although satellite-based observation is indispensable for large-scale survey of MIZs, current satellite payloads and datasets are insufficient for systematic coverage of MIZs in both polar regions. Especially, the lack of a long-term record for the wave-affected MIZs limits both process studies and the detection of potential changes of the MIZs with global warming.

In this study, we utilize ESA's CryoSat-2 satellite (CS2) for the retrieval of wave-affected MIZs, focusing on the region of Atlantic Arctic. Within the Atlantic Arctic, including Barents Sea and the Greenland Sea, there exist a variety of sea ice conditions, such as the young and first-year ice (FYI), as well as the thick, multiyear ice (MYI) advected from the Arctic Basin. Also, frequent storms pass through the sea ice edge during winter, making it a good study area for wave-affected MIZs (Rinke et al., 2017). Besides, the Atlantic Arctic is an important region for human activities, which is highly variant and susceptible to changes with the ongoing Atlantification (Polyakov et al., 2017). In order to study the wave-affected MIZs, we design the retrieval algorithm based on the delay-Doppler radar altimetry, and generate a long-term record for the Atlantic Arctic based on CryoSat-2 for the twelve winters from 2010 to 2022. The paper is organized as follows. In Section 2 we introduce the CS2 dataset and other related datasets that are used in this study, including IS2, SIC, and Sentinel-1 SAR data. Section 3 covers the retrieval algorithm and the analysis of two typical cases of retrieval. Further in Section 4, we compare the MIZ retrieval
with CS2 with that based on IS2 (Horvat et al., 2020) and SAR images (details of the spectral analysis in Sec. B). Section 5 introduces the 12-year record of the wintertime MIZs in the Atlantic Arctic and carries out related analysis. Specifically,





as shown through intercomparisons, the traditional SIC-based MIZ definition yields much narrower MIZs than our retrieval. Finally, in Section 6 we summarise the paper and discuss related topics of the satellite-based observations of the MIZ.

## 2    Data for MIZ retrieval and analysis

### 2.1    CryoSat-2

Since 2010, the CryoSat-2 satellite (CS2) has been constantly observing the earth's cryosphere for over 12 years, and it constitutes one of the most important sources of information for sea ice mass balance (Wingham et al., 2006). The main payload onboard CryoSat-2, SIRAL, is a Ku-band delay-Doppler radar altimeter. Within polar waters, CS2 (or SIRAL) mainly works in SAR or SARIn mode. By utilizing the Doppler frequency shift from consecutive radar signals, we can differentiate the backscatter from different along-track positions of the satellite. As a result, the along-track resolution (or the effective footprint size) is greatly enhanced to about $400m$, much improved from traditional pulse-limited altimeters. Furthermore, besides the traditional gated waveform power, the backscattered radar signals for the same footprint, denoted looks, form a waveform stack that contain extra information of the ocean's surface. Traditionally, CS2's observation over sea ice is mainly utilized for the retrieval of the water level and the sea ice thickness (Meloni et al., 2020). The range retracking, the classification of surface types, the retrieval of the radar freeboard and the conversion into ice thickness are carried out. However, due to the relative coarse resolution of CS2 with respect to the typical wavelengths in MIZs, as well as the range uncertainties (Xu et al., 2020), CS2 has not been applied to the study of MIZs.

The schematics of CS2's observation in the polar ocean is shown in Figure 1, with the satellite's ground track traversing the open ocean, through the MIZ, and into the ice pack. Wind waves affect the ice cover by wave/swell generation, the propagation into the ice edge, and the ensuing interaction with sea ice, including breaking the sea ice into smaller floes and the wave attenuation. Given that the ground speed of CS2 is about 8 $km/s$, we consider that for each satellite pass, CS2 captures the instantaneous status of the underlying MIZ.

CS2 waveforms and waveform stacks from an example track in the Barents Sea are also shown in Figure 1. We further examine the following waveform parameters of CS2 for the MIZ retrieval. First, the beginning location of the MIZ along the track can be detected through the change of the waveform power, mainly due to the difference in the backscatter properties between the ocean water and the sea ice. Even partial coverage of sea ice within the CS2 footprint ($400m$ by $1500m$) can greatly affect the overall backscatter coefficient ($\sigma_0$, in dB). Second, within the wave-affected MIZ, waves and swells modulate the surface topography, and with the gradual wave attenuation in the MIZ, the wave power is more concentrated towards the low-frequency, long-wavelength components (Brouwer et al., 2022; Ardhuin et al., 2017; Horvat et al., 2020). The wave-modulated ice topography in the MIZ mainly has two features: (1) the wave amplitude-related height distribution, which is hightly different from the typical sea ice cover, and (2) the slope of the surface modulated by both wave power and wavelength. Third, on the inner ice pack which is not affected by the wave, the surface topography mainly follows the positively skewed distribution (due to ice thickness distribution), with intermittent, low-lying sea ice leads. On the sea ice, the volume scattering



is also highly variable, with more prominent backscatter on the MYI than FYI, as well as highly reflective at nadir looks for
sea ice leads.

  Therefore, CS2 waveforms on the wave-affected MIZs have the following characteristics. Due to the wave-induced sloping,
for the CS2 waveform stack, the power deviation from different looks (i.e., slant looks) is smaller than that on the sea ice, and
comparable to that on the ocean. This characteristic is directly indicated by the Stack Standard Deviation (SSD) parameter,
which is computed as the standard deviation of the Gaussian fit to the range-integrated waveform stack power (in watts).
Besides, due to the large elevation variability in the MIZ,the trailing edge is much wider than that of the typical waveforms
on the sea ice, which is usually dominated by snow and ice volume scattering [see examples in Rapley (Rapley, 1984)].
The Trailing Edge Shape (TES) parameter of the waveform describes the speed of the power decrease of the look-integrated
waveform after the peak power. Specifically in this study, TES is redefined as the fitted $e$-folding parameters of the waveform
power decay in the waveform's trailing edge between 80% and 5% the highest power: $P(x) = P^* \cdot e^{-\frac{x}{TES}}$, where $x$ is the
gate number, $P(x)$ the waveform power within the specified range of the gates, and $P^*$ and TES the two parameters to be
determined. As shown in Figure 1, while the backscatter is similarly strong on ice-covered regions, the value of the SSD and
TES within the MIZ lies between those on the open-ocean and those on the inner part of the ice cover. For this study, we use
the SSD as provided in ESA's baseline of CS2 (baseline-D for the period before April, 2021, and baseline-E for afterwards).
For the TES parameter, we compute its value for each CS2 waveform.

## 2.2 Auxiliary input datasets

Daily sea ice concentration (SIC) maps are usually generated with passive microwave imaging payloads, and the continuous
observation dates back to October, 1987 and consititutes one of the longest record of sea ice. For MIZ studies, in Strong and
Rigor (2013) the region with SIC between 15% and 80% is used as the proxy for the MIZ. In this research, for the CS2 era, we
utilize the SIC product generated at the University of Bremen, which is mainly based on the payload of Advanced Microwave
Scanning Radiometer 2 (AMSR2) and the ARTIST Sea Ice (ASI) algorithm (Spreen et al., 2008). For the study period without
AMSR2 data (i.e., before 2012), we use the SIC product that is also hosted at the University of Bremen and based on the Special
Sensor Microwave Imager/Sounder (SSMIS). By default, the 6.25 $km$ resolution SIC product is utilized, which is sufficient for
various analyses in this study, including the determination of large-scale sea ice edge, as well as the intercomparison with the
MIZ width defined by SIC.

For the atmospheric and wave conditions during the CS2's observations, we mainly rely on the global ERA5 reanalysis
product (Hersbach et al., 2018). Specifically, hourly sea-surface pressure fields (0.25° resolution) and the wave spectra (0.5°
resolution, defined over regions with SIC<15%) are utilized. Although ERA5 does not include an interactive sea ice component,
its wave product over the ocean is extensively validated with in-situ wave measurements globally (Wang and Wang, 2022). The
wave product is also well validated and further used in various studies of the MIZ and polar oceans (Vichi et al., 2019; Alberello
et al., 2022).



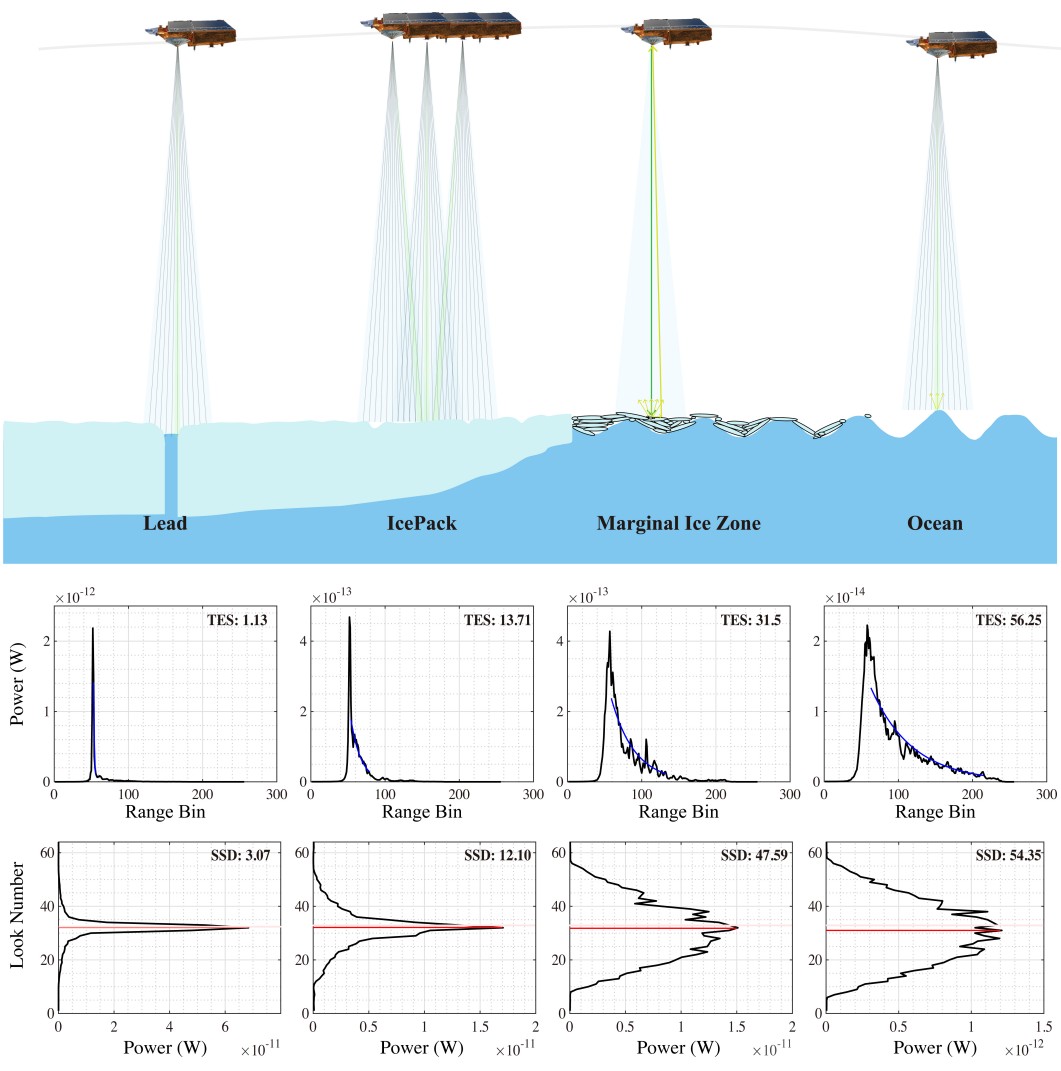

**Figure 1.** CryoSat-2 (CS2) observation of the polar ocean (top panel). CS2 SAR mode waveforms are shown for the 4 typical surface types in lower panels, including open ocean (right column), wave-affected marginal ice zone (MIZ, second to the right), ice floe (second to the left), and sea ice lead (left). The waveforms are chosen from the CS2 track in Fig. 4. The aggregated waveforms (second row) are shown with the exponential fitting of the power decay in the trailing edge and the fitted parameter of the Trailing Edge Shape (TES). Correspondingly, the power of the waveform stack and the waveform Stack Standard Deviation (SSD) are shown in the bottom row.





## 2.3 Other satellites for the MIZ retrieval

### 2.3.1 Sentinel-1

Sentinel-1 (S1) is a polar-orbiting, C-band ($5.4GHz$) Synthetic Aperture Radar (SAR) satellite constellation by the European Space Agency (ESA) and a part of the Copernicus program. The two satellites, Sentinel-1A (launched in April, 2014) and Sentinel-1B (launched in April, 2016) mainly works in the dual-polarization (HH and HV) and Extra-Wide (EW) swath mode in the Arctic region, providing a comprehensive coverage of the Atlantic Arctic. In this study, we mainly use the Ground Range Detected (GRD) product of the EW mode, and the satellites' swath width is about 400 $km$, and the spatial resolution at $40m$. For pre-processing the images, we also apply orbit files, thermal noise correction, radiometric calibrations, and terrain correction and convert the backscatter intensity to decibels (dB) with ESA's Sentinel Application Platform (SNAP).

At $40m$ resolution, only waves/swells with long wavelengths are identifiable, which potentially limits the use of EW SAR images to the cases with strong, deep-penetrating waves and wide MIZs (Brouwer et al., 2022; Ardhuin et al., 2017). For comparison, under the wave mode of S1 satellites ($5m$ resolution), the wave spectra and its components can be better studied (Sutherland and Dumont, 2018; Huang and Li, 2022). For the detection of waves in ice with SAR images, we utilize both visual inspection and the spectral analysis method. Specifically, within the sea ice covered region of the SAR image, we identify wave patterns with interleaving bright/dark stripes of the radar backscatter and reasonable wavelengths. Furthermore, the quantitative spectral analysis is also carried out on local parts of the SAR image (30 $km$ window size), and the spectral peak is identified and associated with the wave in sea ice. In Appendix B we introduce the method in detail, and SAR images that collocate with CS2 tracks are used for the analysis and the validation of the CS2-based retrieval in Section 4.2 and the supplementary material.

### 2.3.2 ICESat2 and the CRYO2ICE campaign

Compared with the CS2 radar altimeter, NASA's ICESat2 (IS2) is a photon-counting laser altimeter which was launched in the autumn of 2018 (Markus et al., 2017). Over the sea ice, the laser altimeter mainly measures the range/height of the snow surface, while the Ku-band radar signals of CS2 usually penetrates a significant part of the snow cover. In order to better evaluate the synergy of the two altimeters for improved snow and ice thickness retrievals (Bagnardi et al., 2021), the CS2 orbit was raised in July, 2020 to attain collocating tracks with IS2. These collocating tracks are available through the CRYO2ICE campaign (http://cryo2ice.org). In the Atlantic Arctic, we attain 21 collocating track pairs between CS2 and IS2 during the two winters (November to April) of 2020-2021 and 2021-2022 (track information in Appendix A).

On the sea ice, the nominal spatial resolution of the beam segments of the ATL07 product for IS2 strong beams (SB) is about 17 $m$ (cross-track) and less than 20 $m$ (along-track). Therefore, IS2 is capable to resolve the long-wavelength swells in the sea ice and identify the MIZ. Specifically, in this study, we apply the MIZ retrieval algorithm in Horvat et al. (2020) to the collocating track pairs, and compare the result with that based on CS2.



## 3 Retrieving wave-affected MIZ with CS2

### 3.1 Retrieval algorithm

Based on the CS2 waveform properties in the polar ocean, we design the following MIZ retrieval algorithm in Figure 2. The
algorithm mainly utilizes two parameters: the backscatter ($\sigma_0$) and the SSD. First, we detect the beginning of the MIZ with $\sigma_0$
through its contrast between the ocean and the sea ice. In particular, we use the in-situ $\sigma_0$ over the ocean and its variability (i.e.,
the standard deviation of $\sigma_0$, denoted SD) to account for the variant ocean condition. When the backscatter is anomalously
high (i.e., over 3·SD), we detect the presence of sea ice, and mark the location as the outer boundary of the MIZ.

Second, among the various waveform parameters, we adopt the SSD as the indicator for the transition from the wave-affected
part (i.e., the MIZ) to the inner ice pack. The determination of the inner boundary of the MIZ is through statistical tests based
on SSD. We search in the along-track direction for the first lead waveform (available from ESA's baseline product), and from
the location of the lead we record the sample-based distribution of SSD over the inner ice pack (100 $km$ in length, containing
over 300 CS2 footprints). The sea ice lead is a flat surface with highly speckle return as observed by CS2. Hence the wave-
affected MIZ can not extend beyond the location of the first lead, and furthermore, the recorded SSD distribution is used as the
benchmark for further determination of the MIZ's inner boundary.

Third, we restart the along-track search from the MIZ's outer boundary. At each step, we advance into the sea ice direc-
tion, and record the SSD distribution around the search point. A statistical test is carried out for comparing the current SSD
distribution and that of the inner part of the ice pack. Specifically, the Kolmogorov-Smirnov test (KS-test) is adopted for the
comparison of the two sample-based distributions. The Null-Hypothesis (NP) is that the two sets of SSD samples follow the
same distribution, and it is rejected at the prescribed significance level of 0.05. For determining the inner boundary of the MIZ,
we stop the along-track search until: (1) the NP of the KS-test is *not* rejected, indicating that the SSD distribution at the current
location is consistent with that of the inner ice pack, or (2) the lead previously recorded is encountered.

The SSD distribution of the local part of the track is based on a prescribed window size of 10 $km$, containing over 30 CS2
footprints. For larger window sizes, more local samples of SSD is included, hence reducing the the potential of Type-II errors
(i.e., premature termination of the search process and the underestimation of the MIZ length/width). However, larger window
sizes inevitably compromise the spatial resolution of the retrieval. Section 3.4 contains the sensitivity study of the window size
and the trade-offs.

It is worth noting that, other waveform (stack) parameters, including TES, are found to be synonymous with SSD. As
shown in Figure 1 and the following typical retrieval scenarios (i.e., Fig. 3 and 4), larger SSD (in MIZ as compared with ice
pack) corresponds to less power drop in the slant looks, which indicates lower sensitivity of backscatter to along-track look
angle. Coincidentally, higher TES is indicative of the slower decay of waveform power with respect to gate (or time), which is
promoted by both larger height variability and the more 'effective' volume scattering typical to the wave-modulated surfaces.
For comparison, the retrieval algorithm as proposed in Rapley (1984) with the pulse-limited altimeter on SEASAT is based on
the (along-track smoothed) Significant Wave Height (SWH), which mainly relied on the leading edge of the waveform. In this
study we choose SSD over TES (or other parameters) mainly due to the larger contrast of SSD between MIZ and the ice pack



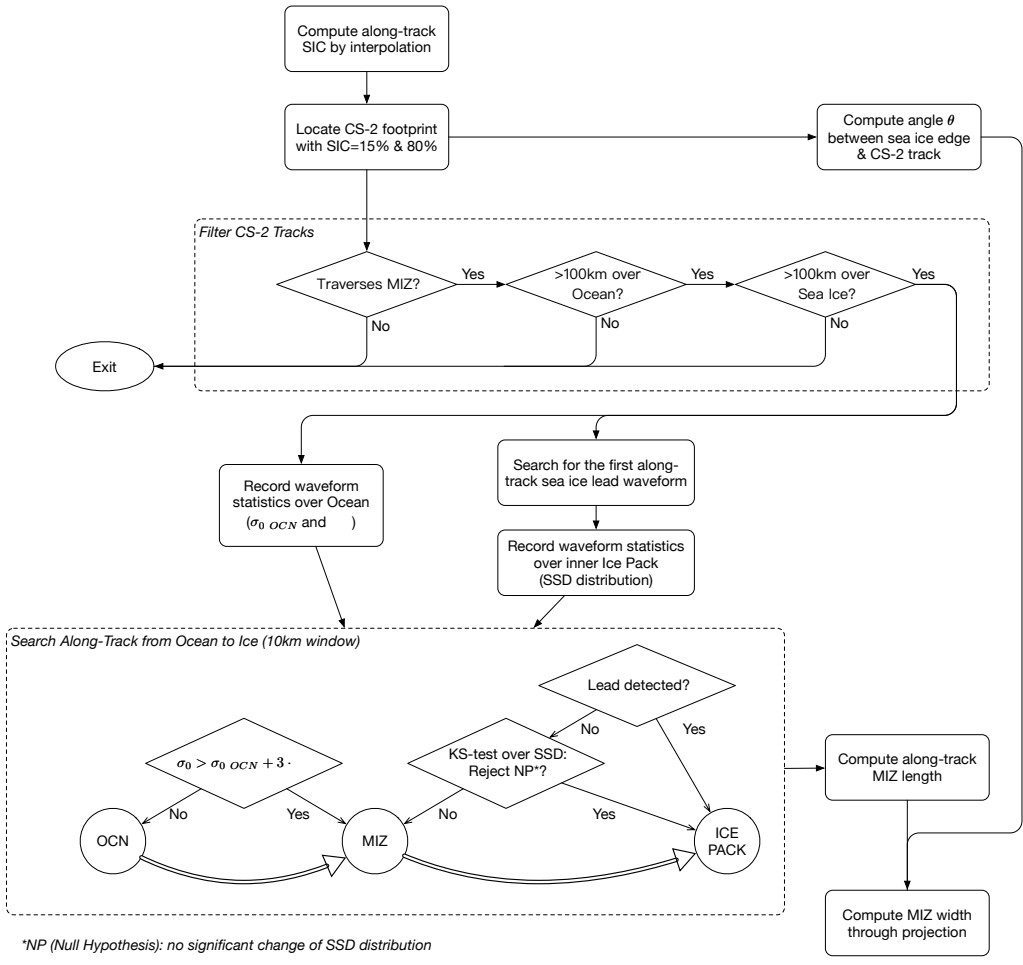

**Figure 2.** Flow chart of the retrieval algorithm.

(regarding their respective variability). To summarise, the proposed algorithm based on SSD has the following advantages: (1) the multi-look capability of CS2 over traditional pulse-limited altimeters; (2) the much enhanced along-track resolution of about $400\ m$ with delay-Doppler treatments; (3) the higher sensitivity for MIZ retrieval with SSD than TES or other waveform parameters. Other retrieval options for historical and future radar altimetry campaigns are further discussed in Section 6.

Fourth, after the inner and outer boundary of the MIZ is determined, we compute the along-track length of the MIZ and compute the MIZ width by projecting onto the normal direction of the local sea ice edge. The determination of the projection angle is based on the sea ice concentration (SIC) maps and introduced below. The projection process is introduced to accommodate the sampling of the CS2 satellite, because there exist arbitrary intersection angles of its ground tracks and the local sea ice edge in the Atlantic Arctic region.





## 3.2 Projection and the computation of MIZ width

In order to determine the intersection angle of the CS2 ground track and the local sea ice edge, we need two directions: (1) the ground track's direction which is readily available from the CS2 product, and (2) that of the local sea ice edge, denote $\xi$. For each MIZ-traversing CS2 track, the daily SIC map corresponding to the CS2's visit time is used to determine the value of $\xi$. Specifically, we first attain all locations with SIC>15% that are adjacent (i.e., within $100 km$) to the ground track's entry point into the ice pack. Second, we carry out the scanning of in the whole range of $\xi$ (from 0 to $\pi$, relative to the east). For each value of $\xi$, a local intersection line is constructed that separates the aforementioned local area into two parts, and the accumulated sea ice extent (SIE) is computed for both sides of the intersection line. Lastly, $\xi$ is defined as the angle under which the SIE difference of the two sides is maximized. The method above, including its parameters, is designed to accommodate: (1) the inherent fractal characteristics of the sea ice edge, and (2) the resolution limitation of the SIC product.

With $\xi$ and the CS2 track direction, we compute the angle of $\theta$, which is the intersection angle for the projection. The width of the MIZ, $W_{MIZ-CS2}$, is then computed as: $W_{MIZ-CS2} = L_{MIZ-CS2} \cdot \sin(\theta)$, where $L_{MIZ-CS2}$ is the along-track length of the MIZ retrieved from CS2. The value $\theta$ in the Atlantic Arctic region is generally larger than $45°$ (Fig. S1), mainly thanks to the high inclination angle of CS2's orbit at $92°$. However, in the Greenland sea, there exist 25% cases with $\theta$ smaller than $30°$. For smaller values of $\theta$, the projection process will incur higher uncertainty in the MIZ width, as further discussed in Section 3.4.

## 3.3 Typical scenarios

We investigate two typical scenarios of MIZ retrieval with CS2. On 2015-Feb-14 (Fig. 3), a CS2 track traversed the sea ice edge in the Barents Sea, and no storm was present in the region of study. The normal direction to the local sea ice edge is almost meridional. As indicated by the ERA5 reanalysis (Hersbach et al., 2018), the total (swell) SWH is about 1.7 $m$ (1.15 $m$) near the sea ice edge. Based on the daily SIC map (6.25 $km$ resolution, produced at Univeristy of Bremen with AMSR2), we compute the along-track locations with SIC between 15% and 80%. By projecting onto the normal direction of the local sea ice edge, we compute the SIC-based MIZ width of about 20 $km$.

CS2 measured marked increase of waveform power from the ocean to the sea ice at about $76.58°N$, which is considered as the starting location of the MIZ. While TES remained stable over the ocean ($55 \pm 3$), it showed: (1) much larger variability on the sea ice and (2) the overall decrease towards the inner part of the ice pack. The overall smaller TES on sea ice indicates a relatively stronger waveform peak, as well as much faster waveform power decay with respect to time (or gate number). Consistent with the changes in TES, the value of SSD also decreased (from over 50 looks on the ocean to less than 20 looks on the inner ice pack), indicating stronger central looks relative to slant-looking ones in the ice pack. Slight shift in the stack center angle is also present, as a result of the gradual decrease of surface height to the north.

For comparison, in Figure 4 we show the case on 2015-Feb-17 with a heavy storm passing around Svalbard (3 days later than the case in Fig. 3). The same storm event is also recorded during the in-situ campaign of N-ICE2015 [denoted M3 in Graham et al. (2019)]. The total SWH is over 3.9 $m$, with the swell power consists over 94% of the total power. The CS2



track entered the sea ice cover at $76.6°N$, and the waveform parameters all showed a gradual transition over a long distance to the relatively calm ice pack in the north. Within the MIZ, both SSD and TES shows not only gradual decrease, but also

large spatial variability than both the ocean and the inner part of the ice pack. The sharp contrast of waveforms in the MIZ to those on the ocean or the inner ice pack is also evident in the overall waveform profile (bottom panel of Fig. 4). Based on SSD and the retrieval algorithm in Section 3.1, we determine that the along-track MIZ terminates at about $79.1°N$. The retrieved along-track MIZ length is over $270\ km$ (yellow shading in Fig. 4). The CS2 observed MIZ length is much larger than that based on along-track SIC (purple shading), which is only $35\ km$.

The nearest available SAR image from Sentinel-1 (Extra-Wide swath mode, $40\text{-}m$ resolution) is 3.1 hours after CS2's observation (Fig. 5). The time difference is within the typical temporal scale of MIZs of 6 hours, hence good collocation between the two satellites (Brouwer et al., 2022). Swells in the ice pack are evident from the SAR image, with the apparent wavelength of about $400m$. Based on the SAR images, MIZ is identified by the outstanding peak of the spectrum of the local backscatter map, with consistent estimation of the wavelength (i.e., Fig. 5.d and e). The intersection angle of the dominant

swell propagation direction and CS2 ground track is about $47°$. As shown in Figure 5.c, the spectral peak that corresponds to the wave structure diminishes to the north of the retrieved MIZ. The CS2 retrieved MIZ termination location is off from that based on the spectral analysis by less than $10\ km$ (4% of the along-track MIZ length). Given the 3-hour difference between the two satellites' visit times, we consider that the CS2 retrieval of the wave-affected MIZ is consistent with that based on SAR images.

Interestingly, the stack center angle of CS2 shows an oscillatory pattern towards the northern end of the MIZ at $79°N$ (Fig. 4.d). The central look (with a Gaussian fitting) is off from the nominal location by as much as $1600\ m$ from the nadir location in the along-track direction. Similar phenomenon is witnessed for many stormy events (another example in Fig. 6). The apparent wavelength of this oscillatory pattern is on the order of kilometers, which is much larger than the swell wavelength (Fig. 5). According to the CS2 dataset, the aircraft yawing and/or pitching is not the main cause. We conjecture it an aliasing effect,

caused by both long-wavelength swells and the misalignment of their propagation direction to the CS2 track.

### 3.4    Sensitivity of retrieval to algorithm parameters

We consider the uncertainty of the retrieval caused by two key parameters: (1) the window size for accumulating the statistics of SSD, and (2) the intersection angle of $\theta$ for the projection. We first evaluate the effect of window size on the retrieved lengths of the MIZ in the along-track direction. Other than the default window size of $10km$, we test two extra window sizes: $5km$

(or 15 CS2 footprints) and $20km$ (or 60 CS2 footprints). With larger window sizes, we generally attain larger values of the MIZ width (Fig. S2). Since more SSD samples are available with larger windows, the false rejection of the Null Hypothesis is reduced during the KS-test (Fig. 2), resulting in wider MIZs. However, the retrieval results with $10km$ and $20\ km$ window sizes are highly consistent, with the correlation coefficient at 0.99, the fitting slope at 1, and only $1\text{-}km$ difference in the MIZ length. Also, at larger window sizes, the spatial resolution of the retrieved MIZ is potentially compromised. Therefore, we

choose the window size of $10km$ by default for all the retrieval studies.

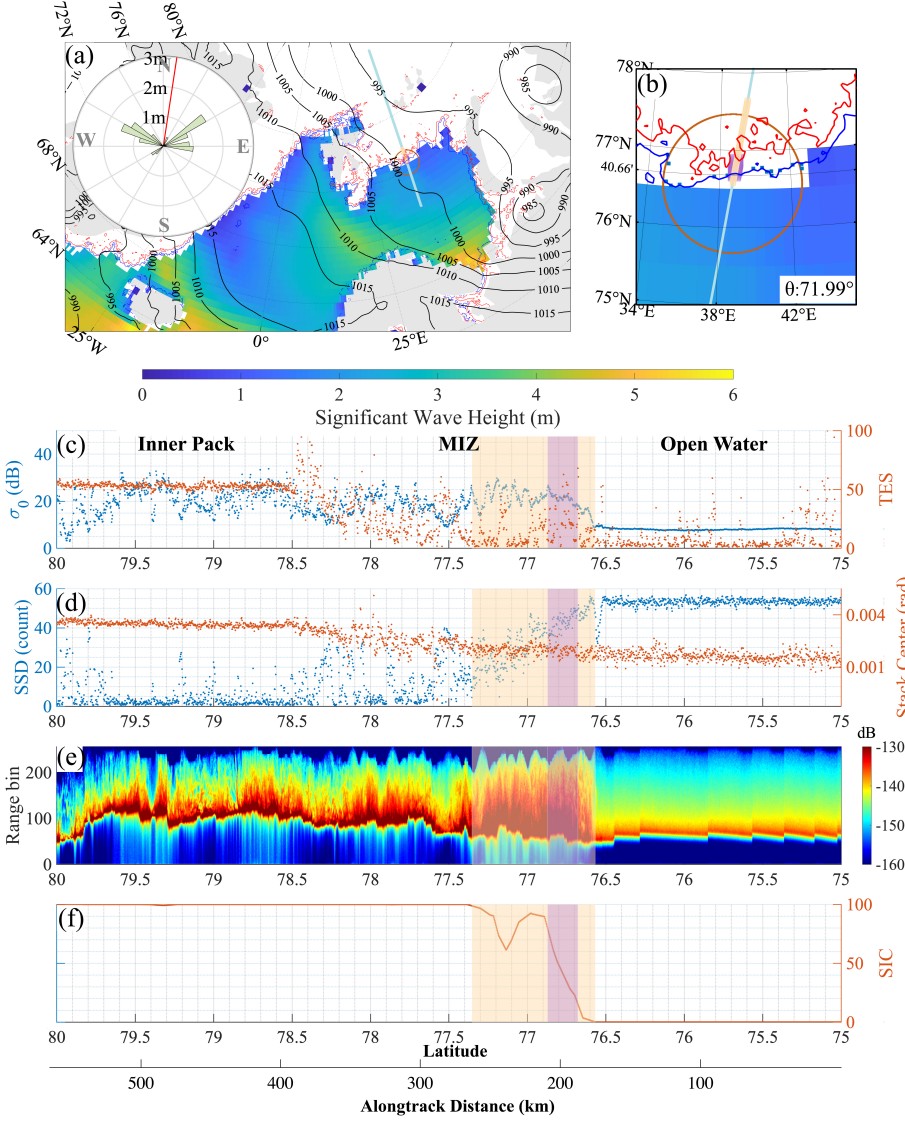

**Figure 3.** CS2 observation of MIZ in the Barents Sea on 2015-Feb-14 (at 00:06 UTC). In the top panels (a and b), hourly total SWH (filled contour) and sea-level pressure (labeled contour lines) from ERA5 are shown with SIC=15% (blue contour line) and 80% (red contour line). CS2 track is shown by the thin light blue line, with the SIC-based (or CS2-retrieved) MIZ highlighted by thick purple (or yellow) line. The inlet rose map shows the swell power/direction spectra near the entry point of the CS2 track into the ice pack (within the circle in panel b), as well as the normal direction into the sea ice edge (red line, details in Sec. 3.2). The intersection angle ($\theta$) between the sea ice edge and the CS2 track is shown in the zoom-in view (i.e., panel b). Along-track CS2 waveform and waveform stack parameters are shown in lower panels, including: (1) backscatter ($\sigma_0$) and TES in panel c; (2) SSD and stack center angle in panel d; (3) the waveform power in panel e; and (4) the along-track SIC in panel e. In lower panels (c to f), the SIC and CS2 retrieved MIZs are also marked with the same colors as in the top panels.



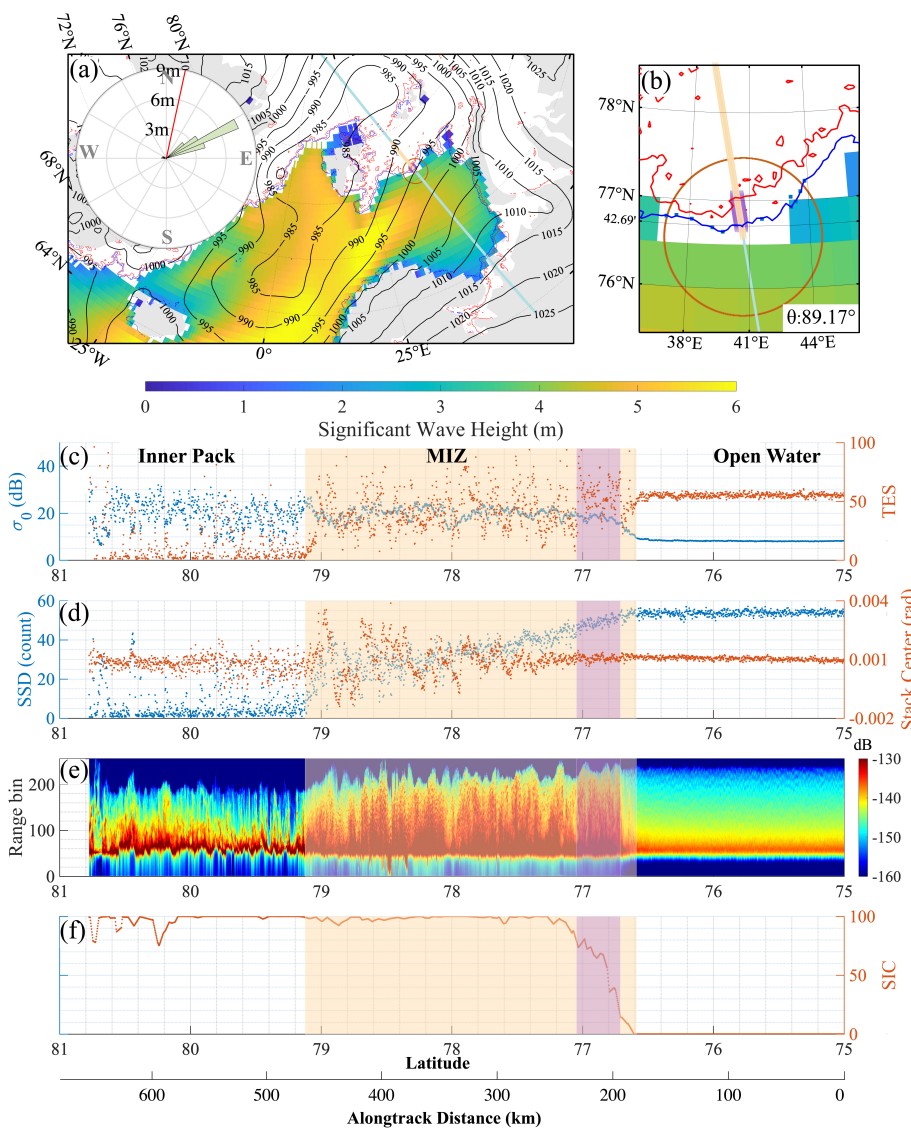

**Figure 4.** CS2 observation of MIZ in Barents Sea on 2015-Feb-17 (at 10:41 UTC). The layout is the same as Fig. 3. Contrary to three days earlier (i.e., Fig. 3), a strong storm is present with waves/swells propagating far into the ice pack.



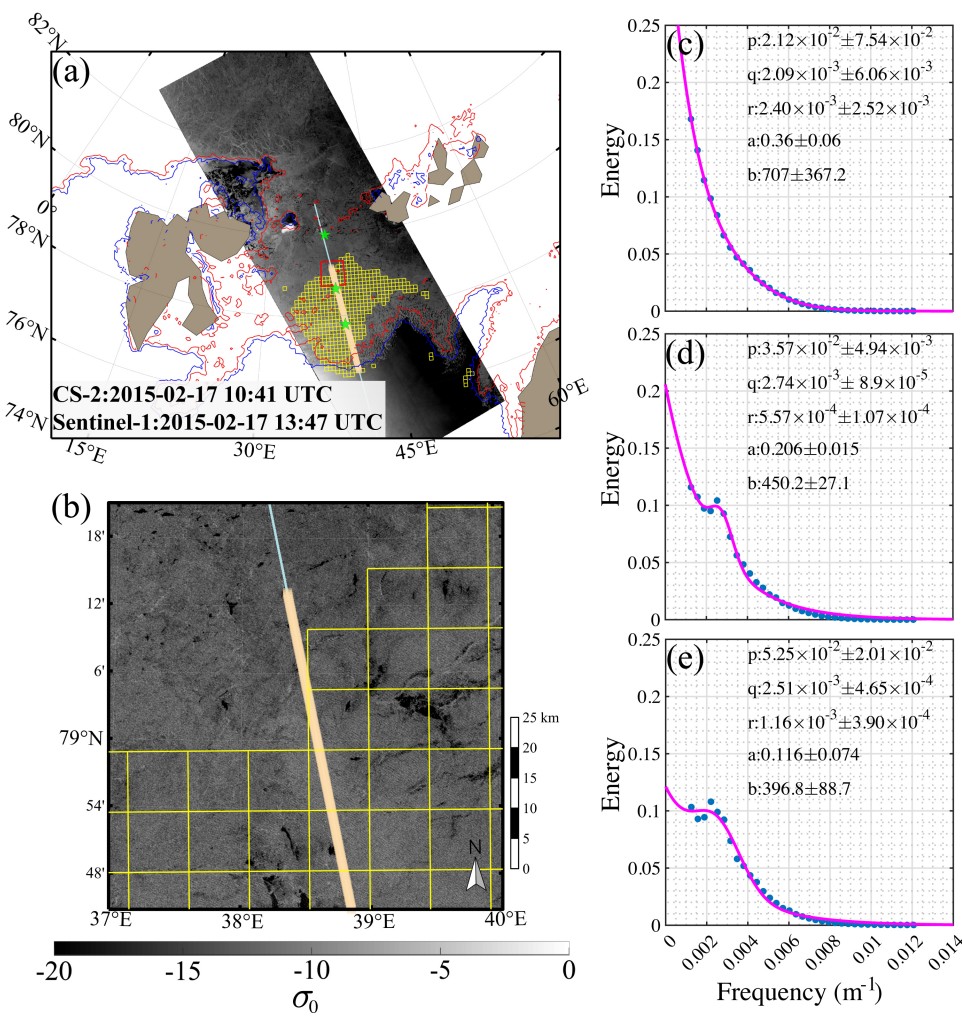

**Figure 5.** Collocating SAR images from Sentinel-1 (EW mode, panel a) for the MIZ in Fig. 4 and the northern end (red box in panel a) of the CS2-retrieved MIZ shown in detail (panel b). The region with detected wave-in-ice by spectral analysis (Appendix B) on the SAR image is marked by yellow boxes ($10km$ scale). The spectra of the Sentinel-1 backscatter map of three typical regions (green dots in panel a) are shown on the right, along with the respective fitted parameters and their uncertainties in Eqs. B1.





We also estimate the relative uncertainty in the MIZ width that is incurred by that in $\theta$. The uncertainty of $\theta$ originates from the sea ice concentration map around the entrance of the CS2's ground track into the ice edge. Through perturbation analysis, we estimate that the uncertainty, denoted $\Delta_\theta$, is on average $6.5°$ in the Atlantic Arctic region. The relative uncertainty of $L_{MIZ}$ due to $\theta$, under the small-angle assumptions, is then computed as:

$$\frac{\Delta_\theta \cdot \frac{dL_{MIZ}}{d\theta}}{L_{MIZ}} = \Delta_\theta \cdot \cot\theta. \tag{1}$$

Among all the tracks, the majority of $\theta$ is larger than $30°$ (e.g., Fig. S1), the relative uncertainty is lower than 20%. Furthermore, in order to ensure 10% or lower relative uncertainty, the value of $\theta$ should be larger than $45°$. For the BS, the NS and the GS region, 88%, 82% and 37% tracks satisfy this criterion, respectively. For satellites with different orbit inclination angles than CS2, the distribution of $\theta$ is different and potentially complementary to that of CS2, especially in the GS region.

## 4 Validation of MIZ observations by other satellites

### 4.1 Validation with ICESat2 from CRYO2ICE campaign

Based on collocating tracks between CS2 and IS2 from the CRYO2ICE program (Bagnardi et al., 2021), we compare the along-track MIZ lengths retrieved with the two satellites. We limit the analysis to the track pairs with the distance between the ground tracks less than $50\ km$. This in effect eliminates the track pairs without actual collocation in the Atlantic Arctic region. Besides, given the highly variant conditions of MIZ, we only study the track pairs with the observation time difference less than 3 hours. Finally, we attain 21 track pairs in the Atlantic Arctic for the two winters of 2020-2021 and 2021-2022 (track information in Tab. A1). For each track pair, we retrieve the MIZ's boundaries with HC20 and the strong beams (SB) in the ATL07 dataset of IS2 (release 5).

In Figure 6, we show an example of MIZ affected by a storm in the Barents Sea, which is observed by a pair of collocating tracks of CS2 and IS2. The two satellites' visit time is separated by 3 hours. Strong swells (swell SWH=$1.95m$, and the total SWH=$2.62m$) propagated into the ice pack, with the CS2 observed MIZ length over $170\ km$. Different from the case in Figure 4, the SIC-based MIZ is comparable to that based on CS2, mainly due to a wide and loose ice edge. For IS2, the observation of MIZ mainly relies on the high-resolution, high-precision elevation measurements over sea ice, which allows direct sampling of waves with relatively long wavelengths (Horvat et al., 2020; Brouwer et al., 2022). The surface elevation measurement in ATL07 product of IS2 only contains valid photon segments over sea ice (i.e., no data on ocean, last panel in Fig. 6). The large oscillatory, wave-like structure of the surface elevation (i.e., periodic signals with amplitude over $50cm$) is evident, indicating the wave-affected MIZ. The gradual decrease of the wave amplitude towards the north implies the wave attenuation within the MIZ. We retrieve the northern end of the MIZ with the algorithm proposed in Horvat et al. (2020) (denoted by HC20 hereinafter). The location of the MIZ's northern boundary as retrieved by IS2 is offset from the CS2 retrieval by only about $1km$ ($< 1\%$ of the total MIZ length). Since the ATL07 product only include valid measurements on sea ice, we treat the south-most photon segment with a valid elevation in ATL07 as the MIZ's southern end observed by IS2. It is worth to note



that, for this specific case, the photon segments are not continuous near the MIZ's southern end, probably due to: (1) the cloud contamination and/or (2) the relatively fine footprints of IS2. In general, we consider the CS2 and the IS2 retrieval of MIZ consistent, especially given the fast changing nature of MIZ and the 3-hour difference of the visit times.

Similar to the case in Figure 4 and 5, we also carry out spectral analysis of the case in Figure 6 (results shown in Fig. S8). The visit time of Sentinel-1 is about 2.5 hours ahead of IS2, and 5.5 hours head of CS2. The apparent wave structure on the SAR image covers over $150 km$ into the ice pack and terminates at $78°N$, which is also well captured by the spectral analysis. The location of wave's presence in the sea ice is highly consistent among the three satellites (all within 10 $km$).

Using all the 21 collocating tracks from CRYO2ICE campaign, we compare the location of the retrieved MIZs by CS2 and
IS2 (the nearest SB to the respective CS2 track). The MIZs' southern boundaries and the northern boundaries are shown in Figure 7 (left and middle panel, respectively). Specifically, we compare the latitudes of the boundaries, given that these tracks are almost meridional in this region. As shown, very high statistical correlations (Pearson's $r$ over 0.99) are attained for both the southern and the northern boundaries of the MIZs. Furthermore, for the along-track MIZ length (right panel of Fig. 7), the retrieval with CS2 and that based on IS2 are also highly consistent ($r$=0.86). The linear regression between CS2 and IS2 yields
a fitting slope of 0.87±0.25, indicating that there is no systematic difference between the two. Besides, the correlation is higher in the Barents Sea than in Greenland Sea, which may be due to a more mobile and spatially non-continuous sea ice cover in the latter. The along-track MIZ length is in the range of 5 $km$ and 180 $km$, indicating that various MIZ conditions are covered, including both calm cases and stormy ones which are associated with wide MIZs (e.g., Fig. 6).

It is worth to note that MIZ retrieval with CS2 and IS2 are based on different approaches. For IS2 the retrieval relies on
the direct observation of wave structures by high-resolution sampling of photon segments. Rather than directly resolving the waves, the retrieval with CS2 is mainly based on the aggregate behavior of radar waveforms over the wave-modulated sea ice cover. One common characteristics to both CS2- and IS2-based MIZ retrieval is that the spatial representation of altimeter is inherently limited. Related issues including the quantification of representation uncertainty are further discussed in Section 6.

## 4.2 Analysis with collocating Sentinel-1 images

Based on the 21 collocating tracks from the CRYO2ICE campaign, we further find available collocating Sentinel-1 images (EW mode). In order to ensure temporal collocation, we limit the observation time of the Sentinel-1 satellites to be within 6 hours of that by CS2. In total, there are 9 cases with the collocating observation of all three satellites (an example in Fig. 6). We carry out visual inspection as well as spectral analysis for all the SAR images, and the results are listed in Table 1 and supplementary figures (Fig. S3 to S11).

Among the 9 cases, there are 6 with evident wave penetration in the sea ice cover. The spectral analysis successfully identifies 4 out of the 6 cases, with good consistency among the retrieval results by CS2, IS2 and S1 (case #2 #3 #4 and #6). Case #5 (Fig. S7) features an inhomogeneous ice edge and the mixture of ice floes and open water. Although the visual inspection reveals evident wave structures over the ice covered region, the spectral analysis fails to detect any outstanding peak in the spectrum. Also, for case #9 (Fig. S11), the MIZ detected by CS2 is further to the north of both the IS2 retrieval and the spectral analysis
based on the S1 image.





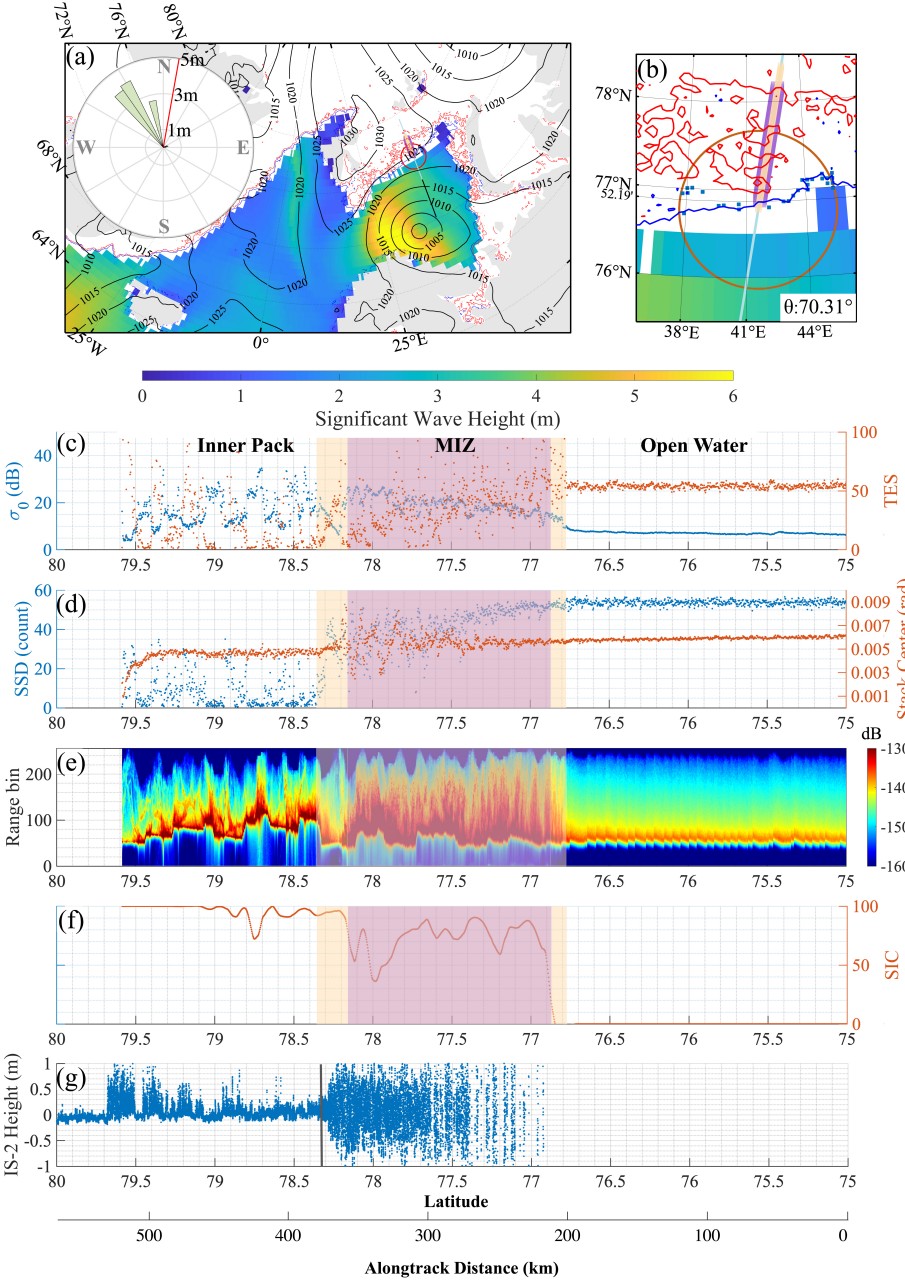

**Figure 6.** CS2 and IS observation of the MIZ in Barents Sea on 2021-Mar-17. Similar to Fig. 4, strong swells propagate into the ice pack, with the MIZ width over 170 $km$. The MIZ is sampled by a pair of collocating tracks by CS2 (at 09:40) and IS2 (at 06:40), with the time difference of 3 hours. The added panel f shows the along-track IS2 elevation, as well as the retrieved northern boundary of the MIZ with HC20 (black vertical line around $78.4°N$).





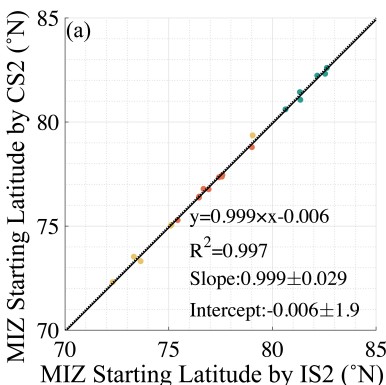
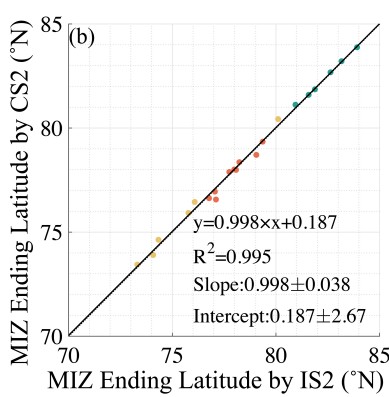
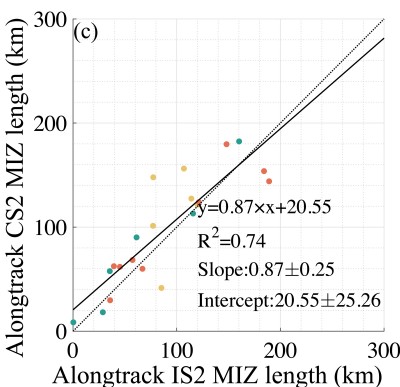

**Figure 7.** Comparison of along-track MIZ retrievals with collocating tracks of CS2 and IS2 in Atlantic Arctic during the winters of 2020-2021 and 2021-2022. Each dot represents a track pair, with 21 pairs in total. The dots are color-coded according to the track locations: orange for those in Barents Sea, yellow for those in Greenland Sea and green for other tracks around Svalbard. The locations are same as Fig. 8. The comparison of the along-track MIZ starting and stopping latitudes are shown (left and middle panel respectively), with that for along-track MIZ lengths (right panel). The linear regression line (solid, black) and the fitting parameters are shown in each panel, together with the 1:1 line (dotted, black).

For the other 4 cases without any waves detected (by either visual inspection or spectral analysis), the dominating processes are from the ocean. For example, for case #1, the frazil streaks governed by new ice formation and Langmuir circulation forms the MIZ and it is successfully identified by both CS2 and IS2. For the ocean turbulence dominated ice edges (i.e., case #4, #7 and #8), the regions with sea ice free drift are also corrected retrieved by both altimeters. The reason why spectral analysis
fails to identify waves for these cases may be due to the coarse resolution of the S1 EW image ($40m$ resolution), as well as the complex, inhomogeneous the ice edge.

## 5 Wintertime MIZ climate record in the Atlantic Arctic

Based on the retrieval for the wintertime CS2 observations, in this section we report the climate record of MIZ in the Atlantic Arctic region for the years from 2010 to 2022. We divide the Atlantic Arctic into 3 sub-regions: Barents Sea (BS, south of
$80°N$ and east of $15°E$), north and northwest of Svalbard (NS, region east of $0°E$ except BS), and Greenland Sea (GS, $30°W$ to $0°W$). In total, there are 2818, 3007 and 3160 valid CS2 tracks for BS, NS and GS, respectively. Temporally, we investigate both the whole winter and the two periods of the winter: the first half from November to January, and the second half from February to April. In Section 5.1 we report the basic statistics of the retrieve MIZ width, and in Section 5.2 its inter-annual variability and the study of typical winters. Finally in Section 5.3 we compare the CS2-based retrieval with the traditional
definition of MIZ based on SIC.



## 5.1 Statistics of MIZ widths

In Table 2 we show the general statistics of MIZ width (i.e., $W_{MIZ}$) of all the 12 winters, and in Figure 8 for every 3-month period. MIZ width follows a skewed distribution in all regions, with the mean width of 78.55 $km$, 41.03 $km$ and 55.98 $km$ in BS, NS and GS, respectively. The modal MIZ widths, which are representative of the typical, non-stormy conditions, are: 32.04

$km$ (BS), 11.20 $km$ (NS), and 39.53 $km$ (GS) respectively. Correspondingly, the distribution of $W_{MIZ}$ is highly skewed, and the cases of wide MIZs associated with storm events (examples in Fig. 4 and 6).

Among the three regions, the widest MIZs manifest in BS, with the largest width reaching over 250 $km$ in most winters. Also, within each winter of the BS region, the MIZ width generally decreases in the later stage of the winter. This phenomenon is not observed for the other two regions. The potential reason may be due to ice thickening as the winter progresses, which is

more evident in BS.

In NS, the MIZ is generally narrower than in BS and GS. Especially, for certain years (such as 2014-2015), the sea ice edge is only present to the west of Svalbard (i.e., no ice edge north of Svalbard). Sea ice in NS mainly originates from within the Arctic Ocean, due to the ice advection through the transpolar drift and the interaction with the Atlantic inflow. Consequently, the swell's penetration into the ice pack is potentially limited, and the MIZ is generally narrower in NS.

Among the three regions, GS shows overall the largest modal MIZ widths. On the other hand, the mean MIZ width is smaller in GS than BS, mainly due to the extremely wide MIZs which are more common in BS. Coincidentally, the skewness of the MIZ width distribution is also the lowest in GS. This is mainly due to the generally loose ice pack in GS, as a result of the south-bound, fast ice drift and divergence.

During the study period from 2010 to 2022, we do not observe statistically significant change in the wintertime MIZ width.

Similarly, there is no significant change in extreme cases of MIZ width (i.e., top 5%) for the three regions of the Atlantic Arctic. For comparison, no significant change of SIC-based MIZ width is observed for the same period (2010 to 2022) either, despite that it is generally much lower than the CS2-based retrieval.

## 5.2 Inter-annual variability and typical winters

Although no change in MIZ width is detected, there exists large temporal variability, both inter-annually and intra-seasonally.

In particular, in Figure 9 we show that there is pronounced inter-annual variability (IAV, 2-year cycle) of the extreme MIZ widths (top 10%) in the Barents Sea. For comparison, the modal width in the Barents Sea (e.g., non-stormy condition) does not show similar variability. Collaterally, the mean width shows similarly pronounced IAV, caused by the cases with extremely large widths.

The extremely wide MIZs are caused by various factors, including strong storm events, relatively thinner/looser ice edges,

etc. We would like to note that, in the Barents Sea, the IAV of the widest MIZs coincides with the statistically significant correlation of seasonal mean MIZ widths between the CS2-based retrieval and that based on SIC (details in Sec. 5.3). For winters with a relatively loosely packed ice edge in the Barents Sea, the SIC-based MIZs tend to be wider, and the ice edge is also more susceptible to storms and wave intrusion. However, the quantitative role of these contributing factors, including



**Figure 8.** Statistics of wintertime MIZ width from 2010 to 2022. Two 3-month periods of each winter (Nov.-Jan. in blue and Feb.-Apr. in red) are shown for Barents Sea (BS, top panel), north/northwest of Svalbard (NS, middle panel) and Greenland Sea (GS, bottom panel), separately. The median, the inter-quantiles (box), and the 5th and the 95th percentiles (vertical line) of MIZ width distribution are shown. Statistics of SIC-based MIZ width on the same CS2 tracks are shown in lighter colors.

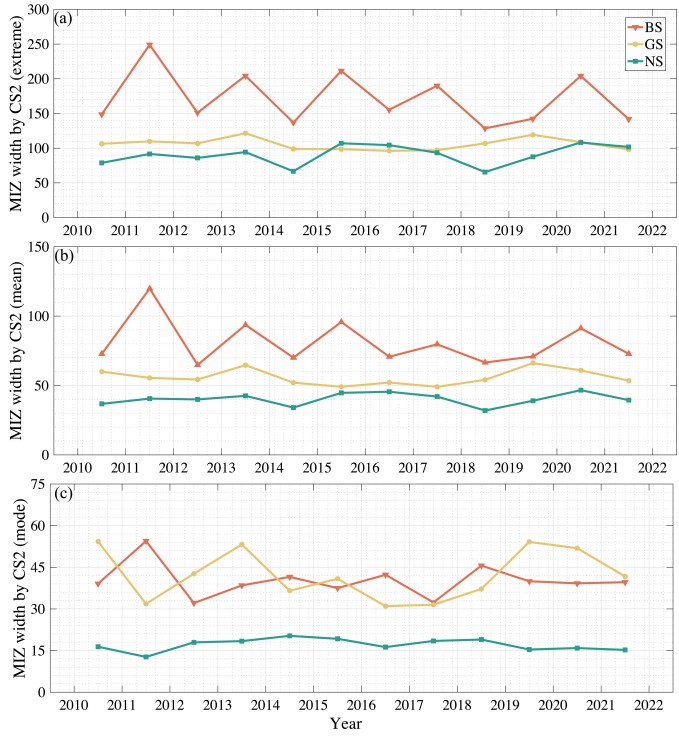

**Figure 9.** The extreme (a), the mean (b), and the modal MIZ widths (c, in $km$) of each winter from 2010 to 2022. Specifically, the extreme MIZ width is computed as the mean MIZ width of the widest 10% MIZs of each winter. Note the difference in the MIZ width ranges (from 300 $km$ in panel a to 75 $km$ in panel c).

the IAV of storms and the ice thickness, is beyond the scope of this study and planned for future work. For comparison, in the

other two regions (BS and NS), we have much lower inter-annual variability in the extreme MIZ widths than the Barents Sea.

Due to the large IAV of the MIZ width, we examine in detail two winters for comparative study: 2012-2013 and 2014-2015, the results shown in Figure 10 . The winter of 2012-2013 followed the record minimum of Arctic sea ice extent in September, 2012. Besides, it was a relatively calm winter in the Atlantic Arctic, with very weak storms throughout the season (Rinke et al., 2017). The sea ice coverage gradually increased in the Barents Sea as the winter progressed from November to January (top

panels of Fig. 10), mainly due to the in-situ ice growth, assisted by the advection from the north. During this period, although only weak storm events were present, yet wave-affected MIZs extended as far as $85°N$ (i.e., 600 $km$ north of Svalbard). For the latter 3-month the 2012-2013 winter, the wave-affected MIZ around Svalbard is not prominent as the former period, only manifesting in the Barents Sea.

Since the sea ice minimum in September, 2012, the Arctic sea ice cover had undergone recovery up to 2015, with both

larger ice coverage and thicker ice (Tilling et al., 2015). Furthermore, the winter of 2014-2015 witnessed frequent storms in the Atlantic Arctic region (Graham et al., 2019). These characteristics are also reflected in the wave-affected MIZs (lower panels in Fig. 10). Contrast to the winter of 2012-2013, there was already large ice coverage in the Barents Sea ($77°N$) since November,

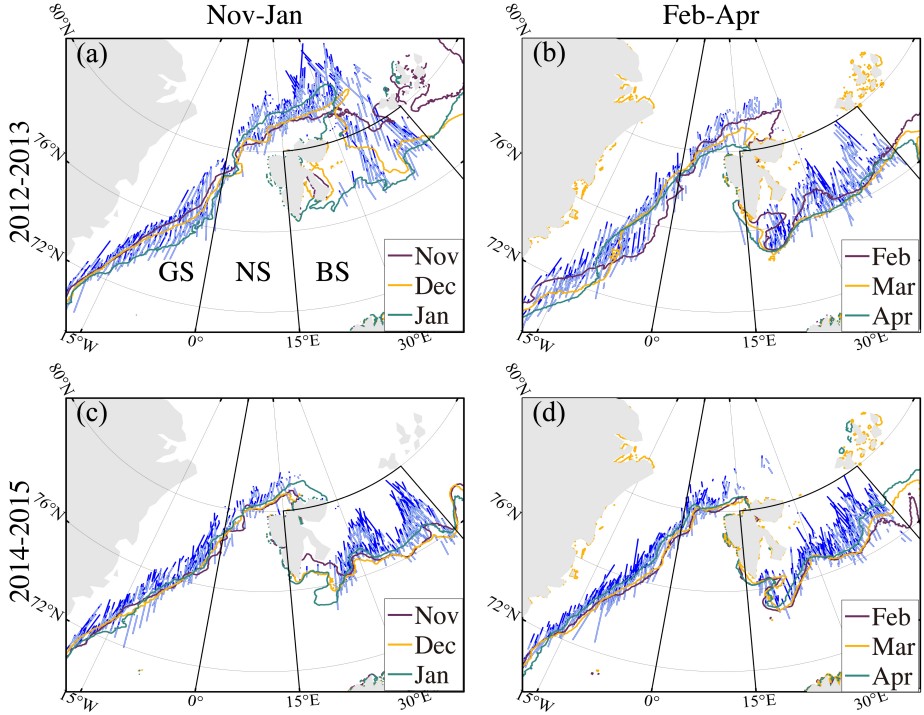

**Figure 10.** Along-track CS2 retrieved MIZ of two typical winters: 2012-2013 (top row) and 2014-2015 (bottom row). The two periods of the winter (Nov.-Jan. and Feb.-Apr.) are shown on the left and right panels, respectively. The monthly mean sea ice edge is shown for each month with contour lines in all panels. For each CS2 track, the part with (daily) along-track SIC lower than 80% are shown in light blue, and those over 80% in dark blue.

2014. The sea ice coverage generally remained high throughout the winter. However, due to frequent storm activities, CS2 observed MIZ extends into the ice pack of over 250 $km$ in both the Barents Sea and the Greenland Sea. Besides, given the larger ice coverage and potentially thicker ice than the winter of 2012-2013, we do not observe any MIZ beyond $82.5°N$ during the whole winter of 2014-2015.

### 5.3 Comparison with SIC-based MIZ

We carry out systematic comparison between the CS2-based MIZ width retrieval and the traditional MIZ definition based on SIC [i.e., SIC between 15% and 80%, as in Strong and Rigor (2013)]. Specifically, two SIC-based MIZ widths are computed. The first method is demonstrated in the examples in Figure 3, 4 and 6, which is based on the SIC along the CS2 track. I.e., for each CS2 track, we attain the along-track SIC and compute the distance between SIC=15% and SIC=80% as the along-track MIZ length. Then the MIZ width is computed with the same projection method as in Section 3.2. The second method is: for each CS2 track, we compute the MIZ width based on the aggregate area with SIC between 15% and 80% in the adjacency

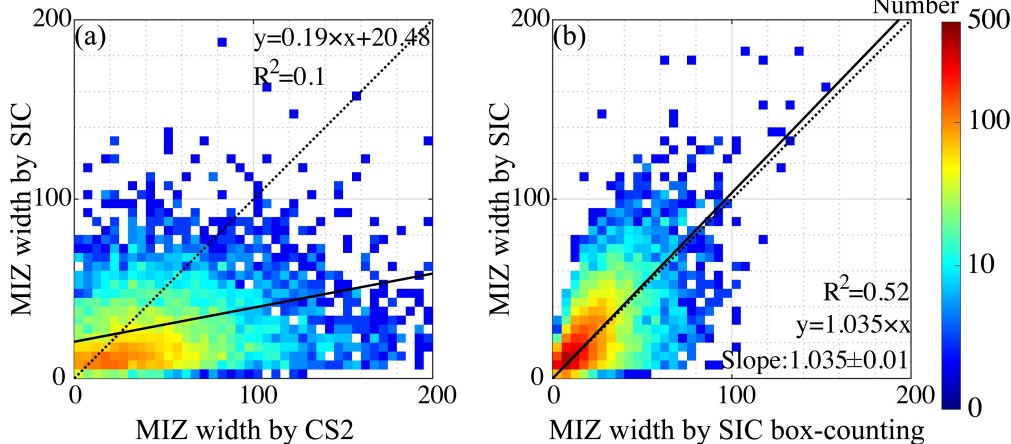

**Figure 11.** Comparison of MIZ width based on CS2 retrieval and the along-track SIC (a), and that of the SIC-based MIZ width retrieved with along-track SIC and the box-counting method (b).

of the track (within 100 $km$ of the track). This method is inherently based on box-counting and it is free from the potential
representation issues with altimetric scans of the MIZ.

Both Table 2 and Figure 8 compares the SIC-based retrievals with the first method. As shown, the SIC-based MIZ width also follows a highly skewed distribution. However, the MIZ defined with SIC is systematically narrower than the CS2 retrieval, including both mean and extreme widths. For example, for BS, GS and NS, the mean width is lower by 43%, 52% and 39%, respectively. More importantly, there is only weak statistical correlation between the SIC and the CS2-based MIZ widths (10%
common variance, Fig. 11.a).

At larger temporal scales (i.e., 3-month), the mean MIZ width based on SIC correlates with that based on CS2 only in the BS region (with the correlation coefficient $r$ of 0.62 and the $p$-value $< 0.01$), but not in the GS or NS regions. For the BS region, the correlation is significant at the monthly and the inter-annual scales ($r$=0.57 and $r$=0.75 respectively, and the $p$-values both lower than 0.05). This statistical relationship may *not* be due to the inherent physical relationship between the wave-affected
MIZ and the daily SIC, but the large-scale sea ice conditions, including ice edge advance and ice thickening throughout the winter.

Between the two SIC-based retrievals, there is overall consistency between two ($R^2 = 0.52$, Fig. 11.b). The box-counting method yields slightly lower MIZ widths (by about 3.5%), and we consider it as a minor issue and due to the practical way for the computation of the area with 15%<SIC<80%. More importantly, the comparison in Figure 11.b reveals the representation
uncertainty with altimetric observations of the MIZs. It is worth to note that, similar to the intercomparison between CS2 and IS2 retrievals (Sec. 4.1), both temporal and spatial representation should be accounted for during the altimetric observations of the MIZ. The representation issue and the potential with the synergy of multiple altimetry campaigns are further discussed in Section 6.





## 5.4 MIZ dataset

We provide the MIZ dataset in both the original along-track format, as well as the gridded version. The gridded dataset is based on the along-track MIZ retrieval results, and it records the presence of the MIZ on the monthly basis. The latitude-longitude grid is adopted, with the spatial resolution of $2°$ in the zonal direction and $1°$ in the meridional direction. Hence, the nominal spatial scale of the dataset is about $100\ km$. For each MIZ-traversing CS2 track of the month, we mark all the grid cells that contain the retrieved MIZ locations along the track. In total, the gridded dataset includes 72 winter months from 2010 to 2022.

The MIZ dataset is available at: https://zenodo.org/record/8176585 (last access: 24 July 2023). The along-track MIZ retrieval result is included for each CS2 track, and the following information is provided: (1) the original CS2 track information; (2) the date (year, month, date) and time (hour) of the CS2 track; (3) the region of the CS2 track (BS, GS or NS); (4) the start location (latitude and longitude) of the retrieved MIZ; (5) the end location (latitude and longitude) of the retrieved MIZ. In total 8985 CS2 tracks are included. The monthly gridded dataset contains 72 NetCDF files, with each file corresponding a single month.

Each file contains the following information/variables: (1) the time; (2) the region flag (i.e., BS, NS or GS); and (3) the MIZ flag (1 for the presence of MIZ within the month, and 0 for the case of no detected MIZ).

## 6   Summary and discussions

In this study we design a new retrieval method for the wave-affected marginal ice zones with the radar altimeter of CryoSat-2. The waveform and the waveform stack parameters of CS2 are utilized to retrieve the along-track locations of the MIZs. Based

on the available CS2 dataset spanning the years of 2010 to 2022, we carry out the retrieval for the winter months in the Atlantic Arctic region. The retrieval is validated with collocating observations of ICESat2 and Sentinel-1. The new dataset contains over 8985 MIZ-traversing CS2 tracks, and yields good spatial and temporal coverage of the MIZs in the Atlantic Arctic (Zhu et al., 2023).

     Based on the new dataset, we investigate the status and potential changes of the wave-affected MIZs in the Atlantic Arctic.

No evident change in the mean MIZ width or the widest MIZs' widths is detected during the period of 2010-2022, but large spatial (region-to-region) and temporal (e.g., inter-annual) variability is present. The three regions of the Atlantic Arctic, distinct in their respective sea ice conditions, show drastically different properties of the MIZs. In the Barents Sea, despite the modal MIZ width of $32\ km$, the wave-affected MIZs can reach over $300\ km$ into the ice pack. In particular, there exists pronounced, 2-year cycle inter-annual variability of the extremely wide MIZs in the Barents Sea. The attribution to storms

and sea ice conditions is planned for future work. The modal MIZ width in the Greenland Sea is in general the largest, and the width distribution shows the lowest skewness. The region around Svalbard contains the overall narrowest MIZs, mainly due to both higher ice concentration and thicker ice. Comparison also indicates that the traditional definition of MIZ based on SIC inherently underestimates the wave-affected MIZ width. More importantly, the (daily) SIC maps are not indicative of the wave-affected MIZs (i.e., no statistically significant correlation).



## 6.1 On the SIC-based MIZ definition

Although the daily SIC maps are not indicative of the wave-affected MIZs, there is statistically significant correlation between the mean MIZ width based on CS2 retrieval and that based on SIC at larger temporal scales. In particular, only in the Barents Sea the mean MIZ width based on SIC correlates with that based on CS2 retrievals, although the former is much narrower by 43%. As analysed in Section 5.3, we conjecture this as the result of large-scale sea ice conditions. During many winters, the sea ice edge advance in the Barents Sea ensures large SIC variability on the monthly or larger scales. New ice forms during the sea ice edge advance, and it is more susceptible to wave/swell's effects due to the low thickness. As a result, a more loosely-packed and mobile ice cover forms, which coincides with a wider MIZ.

Given the large sea ice edge changes throughout the winter, if SIC maps at coarser temporal resolutions were used to generate the MIZ maps (same threshold values of 15% and 80%), we could attain a wider MIZ by SIC. On the other hand, if the SIC variability (instead of mean SIC) is used, we also witness a systematic increase in the retrieved MIZ width. However, with either mean SIC or SIC variability, we cannot directly resolve the wave's effect on MIZ. Similarly Vichi (2022) explored defining MIZ based on SIC variability in the Southern Oceans (SO). In our study of the period between 2010 and 2022, with the ongoing Atlantification, the Barents Sea is similar to Southern Oceans in terms of the ice type, thickness, as well as the seasonal ice edge advance. Based on the analysis above, we consider the SIC at coarser temporal scales is only statistically indicative of the wave-affected MIZs under limited sea ice conditions (i.e., BS, SO). More study is needed to better understand the general applicability of using SIC maps for defining MIZs, especially for future climate changes in the polar regions.

## 6.2 Representation issues for the altimetry-based MIZ observations

Traditional approaches for observing waves in MIZ with satellites are usually based on imaging payloads (Ardhuin et al., 2017; Stopa et al., 2018; Collard et al., 2022). Observing the MIZ with altimeters are inherently limited in terms of the per-pass spatial coverage, which applies to both CryoSat-2 and ICESat2. Although waves and swells are driven by the atmospheric weather systems hence has larger spatial structures, the sea ice cover being affected potentially features larger variability with finer structures. The analysis with along-track SIC retrieval and the comparison with the box-counting method (Sec. 5.3) reveals that there exists no systematic bias, but inherent representation uncertainty of altimetric scans of the MIZ. On the other hand, the temporal representation for observing wave-affected MIZ is also limited, especially for the fast on-set process of the MIZs (Collins III et al., 2015).

We further analyze the representation uncertainty, starting with the spatial representation based on different beams of IS2. On the ground, the three strong beams are about $3.3$-$km$ apart in the cross-track direction. We compute the along-track lengths of the MIZ for each of the strong beams (for all the CS2-IS2 track pairs in Sec. 4.1), and further evaluate the statistical relationship between each pair among the three beams. The common variance of MIZ lengths between the beam pairs is between 91% and 95%. Since the modal MIZ width ($32$ $km$ in the BS region) is much larger than the IS2 beams' separation ($3.3$ $km$ or $6.6$ $km$), the remaining variance of about 7% serves as a lower bound of the spatial representation uncertainty for altimetric sampling. Note that there is 26% unexplained variability between the MIZ lengths of the CS2-IS2 track pairs (Fig. 7),



for which the observation by CS2 and that by IS2 is generally separated by 3 hours. Potential limiting factors of the temporal representation include both the sea ice drift (on the order of $1m/s$ under strong forcings) and the fast changing nature of the MIZs through wave-ice interaction (ice floe breaking, rafting, thermodynamic feedbacks, etc). We relate the drift-induced temporal representation uncertainty to the spatial representation, and estimate the temporal representation uncertainty in along-track MIZ width as 19% for the 3-hour time difference (i.e., 26% minus 7%). Given that there are only 21 track pairs in the analysis, better quantification of the aforementioned representation uncertainty can be carried out with more collocating tracks from the CRYO2ICE campaign in the future. Besides, existing MIZ studies with SAR images usually involves the data analysis with each satellite pass. For the study of MIZs with cross-pass SAR images, the aforementioned temporal representation issues should also be accounted for.

## 6.3 Retrieving the MIZ with radar altimetry campaigns

Given the representation uncertainties due to limited coverage by altimeters, there lies great potential in the synergy of multiple altimetry campaigns for improved observations of the MIZ. The Sentinel-3A and 3B (S3A and S3B for short) both contain the delay-Doppler radar altimeter as CS2, and they have a lower inclination angle of the orbit and cover up to $82°N$. As a result, S3A and S3B provide complementary coverage to CS2 in the Atlantic Arctic, both temporally and spatially. The retrieval algorithm based on SSD and $\sigma_0$ in Section 3.1 can be directly applied to both S3A and S3B. Furthermore, the S3A and S3B ground tracks are expected to include more orthogonal scans for the sea ice edge in the Greenland Sea, which could further reduce the uncertainty caused by the projection process (i.e., Sec. 3.4). Also in Collard et al. (2022), the authors demonstrated the signature of swells with the fully-focused treatment to S3A (Egido and Smith, 2017), and it serves as another important direction in utilizing the delay-Doppler type radar altimeters for observing MIZs with both historical datasets and future campaigns such as CRISTAL (Kern et al., 2020).

Besides SSD, other parameters of CS2 waveforms are also shown to be indicative of the wave-affected MIZ in Section 3.1. For example, the TES parameter reflects the surface elevation variability which is modulated by waves, and it is found to be synonymous with SSD but has lower contrast among the open ocean, the MIZ, and the ice pack. In particular, the retrieval method based on TES resonates with Rapley (1984), in which the wave-in-ice is based on the SWH product generated from the Ku-band pulse-limited altimeter onboard the SEASAT satellite. Our retrieval method can also be adapted for the MIZ retrieval with the existing and historical pulse-limited altimeters, such as SARAL AltiKa (Verron et al., 2015) and ENVISAT (European Space Agency, 2018). However, the effect of altimeter mispointing on the radar waveform should be accounted for (Amarouche et al., 2004). Furthermore, a holistic model of the traditional and delay-Doppler radar altimeter waveforms is needed to better characterize both the ice pack and the wave-affected MIZ. Besides, the historical laser campaign of ICESat (Zwally et al., 2002), although limited in along-track resolution (i.e., the Nyquist wavelength of $350m$), can also be synergized with collocating radar altimetry campaigns to construct the long-term record of MIZs in the polar oceans.





## 7    Code and data availability

CryoSat-2 waveform data are accessed through the PDS system provided by European Space Agency (ESA), available at http://science-pds.cryosat.esa.int/ (last access: 30 August 2022). Daily sea ice concentration maps for the study period of 2010-2022 are hosted at the Institute of Environmental Physics, University of Bremen: https://seaice.uni-bremen.de/data-archive/ (last access: 25 October 2022). ERA-5 hourly atmospheric and wave spectra data are available on the Copernicus Climate Change Service (C3S) Climate Data Store, at: https://cds.climate.copernicus.eu/cdsapp#!/dataset/reanalysis-era5-single-levels?tab=form

(last access: 02 November 2022). The collocating tracks between CS2 and IS2 can be downloaded through the online portal of the CRYO2ICE program at: https://cryo2ice.org/ (last access: 10 January 2023). ICESat-2 ATL07 dataset is available from the National Snow and Ice Data Center: https://n5eil01u.ecs.nsidc.org/DP7/ATLAS/ATL07.005/ (last access: 6 October 2022). Sentinel-1 SAR images are openly accessible through ESA's Sentinel-1 data-hub via: https://scihub.copernicus.eu/dhus/#/home (last access: 29 June 2023).

The CS2-based MIZ product (Zhu et al., 2023) is publicly available at: https://zenodo.org/record/8176585 (last access: 17 July 2023). The dataset contains two parts. First, the CS2 track information and the retrieved beginning and the end locations of the MIZ in the along-track direction of each track. In total 8985 CS2 tracks in the Atlantic Arctic region are included. Second, a monthly gridded dataset is also included, which is based on the along-track retrieval results and records the presence of MIZ within the month. Section 5.4 includes detailed description of the dataset.

The MATLAB codebase for the retrieval of MIZ along a single CS2 track is available at: https://github.com/weixinzhu7/miz_retrieval_cryosat2 (last access: 17 July 2023). The codebase includes the core retrieval algorithm, as well as exemplary CS2 record on 14 February 2015 which is downloaded from the repository above.

## Appendix A:  Collocating tracks between CryoSat-2 and ICESat2 from CRYO2ICE campaign

Table A1 lists all the 21 collocating track pairs from the CRYO2ICE campaign in the Atlantic Arctic during the two winters

of 2020-2021 and 2021-2022. In order to ensure both spatial and temporal collocation, we use the following two criteria for the selection of the track pairs: (1) the starting locations of each track pair is limited to be within 50 $km$ to ensure spatial collocation, and (2) the visit times of each track pair to be within 3 hours.

## Appendix B:  Wave-in-ice detection based on spectral analysis of Sentinel-1 EW images

Sentinel-1 EW mode backscatter images are used for detecting wave structures in the sea ice with the spectral analysis method.

Each of the image is of the resolution of 40 $m$ and the size of 400 $km$ by 400 $km$. In total, 21 images are attained for 9 of the collocating track pairs and the case in Figure 4. These images are further subjected to visual inspections and the following spectral analysis.

For each SAR image, we carry out the analysis on the local window of 30 $km$ by 30 $km$ (or 751 pixels in each direction). The local window is slided with the step size of 10 $km$ in both directions to fully cover the whole SAR image. For the spectral



analysis, first, a two-dimensional Hamming window is applied to the local window. Second, we carry out the two-dimensional Fourier transform on the local wind, and further compute the directional-independent spectrum (wavenumber bin of 0.0003 $m^{-1}$). Third, a band-pass filter is applied for the wavelength between $80m$ and $800m$ which is relevant for the detection of waves.

After we compute the spectrum, we apply the fitting in Eqs. B1 to detect any outstanding spectral peak. In Eqs. B1, $x$ denotes

the wavenumber, $f(x)$ the spectrum. The component of $a \cdot e^{-b \cdot x}$ implies the default spectrum of the red noise of the backscatter map, and that of $p \cdot e^{-\frac{(x-q)^2}{2r^2}}$ corresponds to the spectral peak, and the periodic signal in the image. When the fitted parameter of $p$ is greater than 0 with statistical significance, we detect the periodic signal, and the local window is marked as part of the wave-affected MIZ. Besides, the fitted parameter of $q$ indicates the central wavenumber of the detected wave in sea ice.

$$f(x) = a \cdot e^{-b \cdot x} + p \cdot e^{-\frac{(x-q)^2}{2r^2}} \tag{B1}$$

Figure 5 shows the examples of the spectra in both MIZ and the inner part of the ice pack. The detected spectral peaks in different parts of the MIZ are consistent (pane d and e), with: (1) the wavenumber around $2.6 \times 10^{-3} m^{-1}$, and (2) the decrease of amplitude into the inner part of the MIZ (i.e., decrease in the value of $p$'s), indicating wave attenuation. Beyond the MIZ, we do not detect any spectral peak (panel c). Besides, the MIZ determined with the spectral analysis (i.e., $p$ greater than 0 with statistical significance) is highly consistent with the retrieval with CS2. Other examples of the SAR-based MIZ retrievals are

shown in Fig. S3 to S11.

*Author contributions.* SX conceived the overall retrieval framework. SX and WZ designed and implemented the retrieval algorithm. WZ, SX and SL carried out the overall data processing and analysis. All authors contributed to the writing of the manuscript.

*Competing interests.* The authors declare that they have no competing interests.

*Acknowledgements.* This work is supported by the joint project funded by the National Key R & D Program of China (grant no.: 2022YFE0106700)

and the Norwegian Research Council (grant no.: 328957). SX is also partially supported by the National Science Foundation of China (grant no.: 42030602), the International Partnership Program of Chinese Academy of Sciences (grant no.: 183311KYSB20200015), and the Norwegian Research Council under the TARDIS project (grant no.: 325241).



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





**Table 1.** Analysis of the collocating observation by Sentinel-1 with the track pairs from the CRYO2ICE campaign. The type of the ice edge in each case is determined by visual analysis of the Sentinel-1 image.

| Index | Region | Date | $L_{MIZ}$ by CS2 (km) | $L_{MIZ}$ by IS2 (km) | Total SWH (m) | Type of ice edge | Wave-in-ice? | Figure |
|---|---|---|---|---|---|---|---|---|
| 1 | NS | 09-Nov-2020 | 113.08 | 116.08 | 1.19 | Frazil streaks | No | S3 |
| 2 | NS | 13-Nov-2020 | 182.37 | 157.93 | 2.53 | Wave-affected MIZ | Yes[#] | S4 |
| 3 | NS | 30-Nov-2020 | 57.7 | 75.51 | 2.24 | Wave-affected MIZ | Yes[#] | S5 |
| 4 | GS | 11-Dec-2020 | 41.55 | 49.47 | 2.09 | Eddy/turbulence on ice edge | Yes[#] | S6 |
| 5 | BS | 17-Dec-2020 | 68.42 | 40.78 | 1.90 | Inhomogeneity[+] | Yes | S7 |
| 6 | BS | 17-Mar-2021 | 179.64 | 148.18 | 2.79 | Wave-affected MIZ | Yes[#] | S8 |
| 7 | NS | 21-Dec-2021 | 18.19 | 2.87 | 0.51 | Eddy/turbulence on ice edge | No | S9 |
| 8 | NS | 21-Dec-2021 | 62.43 | 35.1 | 1.05 | Eddy/turbulence on ice edge | No | S10 |
| 9 | GS | 24-Jan-2022 | 147.86 | 58.95 | 3.64 | Wave-affected MIZ | Yes | S11 |

\# wave-in-ice detected by spectral analysis on the backscatter map of S1 EW image (see Appendix B).

\+ inhomogeneous ice edge with the mixture of ice floes and open water, with waves only visible on ice patches.

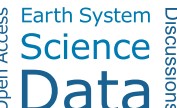

**Table 2.** Statistics of wintertime MIZ width based on CS2 and the along-track SIC during the period of 2010 to 2022.

| $W_{MIZ}$ | CS2 retrieval | | | SIC retrieval | | |
|---|---|---|---|---|---|---|
| Region | BS | NS | GS | BS | NS | GS |
| Mean ($km$) | 78.55 | 41.03 | 55.98 | 44.51 | 25.06 | 26.67 |
| Mode ($km$) | 32.04 | 11.20 | 39.53 | 18.13 | 11.94 | 8.30 |
| Median ($km$) | 58.44 | 29.54 | 47.88 | 32.21 | 18.27 | 19.62 |
| SD ($km$) | 65.21 | 39.95 | 39.39 | 42.24 | 20.42 | 24.38 |
| Skewness | 1.72 | 2.02 | 1.55 | 2.81 | 2.67 | 2.48 |



**Table A1.** Information of the collocating tracks in the Atlantic Arctic from CRYO2ICE.

| Date | CryoSat-2 Track | ICESat2 Track | Region |
|---|---|---|---|
| 9-Nov-2020 | CS_OFFL_SIR_SARI2_20201109T032024_20201109T032943_D001 | ATL07-01_20201109000652_07000901_005_01 | NS |
| 13-Nov-2020 | CS_OFFL_SIR_SARI2_20201113T031635_20201113T032600_D001 | ATL07-01_20201112235833_07610901_005_01 | NS |
| 30-Nov-2020 | CS_OFFL_SIR_SARI2_20201130T035100_20201130T035857_D001 | ATL07-01_20201130003353_10210901_005_01 | NS |
| 2-Dec-2020 | CS_OFFL_SIR_SARI2_20201202T152803_20201202T153045_D001 | ATL07-01_20201202121654_10590901_005_01 | NS |
| 11-Dec-2020 | CS_OFFL_SIR_SARI2_20201211T174754_20201211T175338_D001 | ATL07-01_20201211144312_11980901_005_01 | GS |
| 17-Dec-2020 | CS_OFFL_SIR_SARI2_20201217T010445_20201217T011327_D001 | ATL07-01_20201216220040_12790901_005_01 | BS |
| 27-Dec-2020 | CS_OFFL_SIR_SARI2_20201227T123648_20201227T123803_D001 | ATL07-01_20201227092701_00521001_005_01 | BS |
| 5-Jan-2021 | CS_OFFL_SIR_SARI2_20210105T163428_20210105T164011_D001 | ATL07-01_20210105132734_01921001_005_01 | GS |
| 17-Jan-2021 | CS_OFFL_SIR_SARI2_20210117T130500_20210117T130755_D001 | ATL07-01_20210117095358_03731001_005_01 | BS |
| 26-Jan-2021 | CS_OFFL_SIR_SARI2_20210126T152452_20210126T153941_D001 | ATL07-01_20210126122019_05121001_005_01 | GS |
| 30-Jan-2021 | CS_OFFL_SIR_SARI2_20210130T115637_20210130T120608_D001 | ATL07-01_20210130090326_05711001_005_01 | BS |
| 14-Mar-2021 | CS_OFFL_SIR_SARI2_20210314T201925_20210314T202942_D001 | ATL07-01_20210314172315_12331001_005_01 | BS |
| 17-Mar-2021 | CS_OFFL_SIR_SARI2_20210317T094112_20210317T094316_D001 | ATL07-01_20210317064029_12721001_005_01 | BS |
| 1-Nov-2021 | CS_OFFL_SIR_SARI2_20211101T114343_20211101T115230_E001 | ATL07-01_20211101092023_06101301_005_01 | GS |
| 21-Dec-2021 | CS_OFFL_SIR_SARI2_20211221T073819_20211221T074617_E001 | ATL07-01_20211221051453_13711301_005_01 | NS |
| 21-Dec-2021 | CS_OFFL_SIR_SARI2_20211221T091645_20211221T092544_E001 | ATL07-01_20211221064911_13721301_005_01 | BS |
| 24-Jan-2022 | CS_OFFL_SIR_SARI2_20220124T083803_20220124T084434_E001 | ATL07-01_20220124062532_05041401_005_01 | GS |
| 26-Feb-2022 | CS_OFFL_SIR_SARI2_20220226T040343_20220226T041348_E001 | ATL07-01_20220226014442_10051401_005_01 | BS |
| 28-Feb-2022 | CS_OFFL_SIR_SARI2_20220228T172132_20220228T172443_E001 | ATL07-01_20220228150200_10441401_005_01 | BS |
| 19-Mar-2022 | CS_OFFL_SIR_SARI2_20220319T011701_20220319T012205_E001 | ATL07-01_20220318230308_13241401_005_01 | NS |
| 28-Mar-2022 | CS_OFFL_SIR_SARI2_20220328T051338_20220328T051615_E001 | ATL07-01_20220328030343_00771501_005_01 | GS |