# Peer review of "A 12-Year Climate Record of Wintertime Wave-Affected Marginal Ice Zones in the Atlantic Arctic based on CryoSat-2"

_Earth System Science Data, 2023_

## Referee Comment (RC2)

This is a review of the manuscript entitled "A 12-Year Climate Record of Wintertime Wave-Affected Marginal Ice Zones in the Atlantic Arctic based on CryoSat-2". The manuscript describes a method to retrieve the wave-affected Marginal Ice Zone (MIZ) using Cryosat-2. After introducing the importance of the MIZ, they describe their algorithm to retrieve the inner and outer limits of the MIZ. Then, they apply this algorithm over 2 case studies and discuss their definition of the MIZ against wave detected by Sentinel-1 to evaluate their method. They pursue this evaluation using this time a comparison of their method with ICESat-2 retrieved wave-affected MIZ for collocated tracks, with a special focus on 1 case. Having gained confidence in their algorithms and assessed sources of uncertainty, they extend their study to CS2 tracks in the Atlantic Arctic for the period 2010-2022. They describe the properties of the MIZ in 3 sub-regions and find no significant trend in the MIZ width in any of these regions. Finally, they discuss other sources of uncertainty.

The manuscript is generally well-written and clear. It synthesizes a large amount of work, with a strong emphasis on the validation of the algorithm using a multiple-sensor approach. The science is sound, well-referenced, and the results are well-discussed. Therefore, I recommend this manuscript for publication after minor revisions.

Minor general comments:

I would suggest restructuring section 6 to start with the discussion and end with the summary. I think that would make more sense and conclude the paper on a more "positive" note. I would also suggest concluding by adding a few sentences to give some context to the results. For instance: the dataset is now available to the public and the research community, what type of application do the authors suggest for it? Could we use it already to evaluate the MIZ extent in wave-ice coupled models? What is the next step with this dataset? For instance, is there any plan to retrieve more quantitative data from CS2 in the MIZ (floe size, wave height in ice...)? What is missing to do that? Is there any plan to extend the method to the Pacific Arctic, or Antarctica? Would it work? This conclusion does not need to answer all these questions or to provide an in-depth plan of future work, but I think giving some direction would really improve the impact of the paper.

I have another general comment that is more like a suggestion. The quantity of information lets me think the manuscript could be divided into two: proof of concept one (section 1→4) and a short result article extending section 5. That would certainly increase the impact of section 5 and benefits the authors. Now, the paper is coherent as it is and reads well despite being long, so the decision should be made by the authors.

Specific comments:

L1: "integral part of the ice cover" → I am not sure what this expression means. Important part of the ice cover?
L20: I am not sure "incurred" is the right verb here.

L22: Wave attenuation is a big topic and there is no real consensus on which processes (not all related to friction) dominate depending on wave and sea ice conditions. I would suggest "a diversity of processes"? On this note, I may be a bit biased, but I would suggest that a direct application of this dataset is to gain a better understanding of the processes dominating the wave attenuation by constraining the MIZ extent in wave-ice coupled model (see what Boutin et al. 2022 did with Horvat et al. 2020 dataset for instance).

L23: "more important roles by inducing positive feedback" → Asplin et al. 2012 only suggest it might be the case, but I don't think it has been proven. I would add potentially (by potentially inducing...).

L24/25: The sentence is a bit confusing. Also, I'm a bit picky maybe but I feel "Ingvaldsen et al., 2021" is not the best reference to support the statement made here as it discusses physical and ecological changes, not really changes in human activities.

L32: "and the respective uncertainties" → the phrasing is confusing here. "and are highly uncertain in the MIZ"? (I am sure there must be a reference for that)

L66: "Furthermore, besides [...] that contain extra information of the ocean's surface." I am a bit confused by that sentence. I would recommend splitting it into shorter simpler sentences.

L71: "However, due to the relative coarse resolution of CS2 with respect to the typical wavelengths in MIZs" → Wavelengths is a bit ambiguous here → (surface gravity) wave wavelengths.

L74: "Wind waves affect the ice cover by wave/swell generation, the propagation into the ice edge, and the ensuing interaction with sea ice, including breaking the sea ice into smaller floes and the wave attenuation". This sentence is a bit confusing and needs some rephrasing. (For instance, I understand the first part as "Wind waves affect the ice cover because they can generate swells", which is not correct).

L82 ", waves and swells" → swells are still waves, so maybe "wind waves and swells"?

L83: I feel like these references are not the most appropriate to support the statement made here. The fact that waves get longer as they propagate has been known for a while (I'd suggest Robin, 1963, see below).

L87: on→in ; wave → waves

L91: The authors might want to repeat the reference to Figure 1 at the start of this paragraph, it really helps the reader to look at this figure while reading the description of these quantities.

L107: constitutes

L113: "is utilized" → I think "is used" works better here, and in a lot of places in the rest of the manuscript.

L135: I would recommend referring to a manuscript's figure that shows such patterns (there should be one in Collard et al., 2022 for instance).

L159→165 I find this paragraph confusing, it could be worth re-ordering the information, maybe starting with the introduction of the physical concept (looking for individual leads as a proxy for pack ice), and then explaining how this is done in practice. I would also recommend adding a comment on this choice of defining pack ice with the presence of leads. Technically, the MIZ can be characterized by the presence of many small leads. While I understand the idea of the authors, I think it can be counter-intuitive to potential readers.

L200→203. I find the description of the method to retrieve "xi" hard to follow. I would suggest rewriting it or adding a little schematic.

L215: University

L218: "CS2 measured marked" → I don't understand.

L229: shows → show

L230: "large… than" → "larger .. than"

L248: "is on the order of" → "is of the order of"

Figure 5: Which green points are associated with panels d,e,f?

L329: corrected → correctly?

L354: From the text, I don't understand the reason why the swell penetration is "potentially limited". My guess is that this is because this advected ice is thicker than locally formed one, but this is not clear in the text. Or do the authors mean that there is simply not a large band of ice (and so mechanically a narrow MIZ)? Please clarify.

L424: The gridded product resolution is much coarser than the mean width of the MIZ in the Atlantic Arctic. Is it not a problem? I would recommend justifying this choice and detailing what limits the choice of finer resolutions (e.g., the sampling of CS2?).

References:

Robin, G. de Q. (1963). Ocean waves and pack ice. Polar Record, 11(73), 389–393. https://doi.org/10.1017/S003224740005350X

Boutin, G., Williams, T., Horvat, C., & Brodeau, L. (2022). Modelling the Arctic wave-affected marginal ice zone: A comparison with ICESat-2 observations. Philosophical Transactions of the Royal Society A: Mathematical, Physical and Engineering Sciences, 380(2235), 20210262. https://doi.org/10.1098/rsta.2021.0262

Best regards,

Guillaume Boutin

---

## Author Comment (AC1)

Reply to Review #1:

The authors would like to thank the reviewer for the invaluable comments and suggestions. Below are the replies to each point raised in the review, accompanied by the specific revisions that have been made. The original review comments are presented in *blue italic* font and organized in paragraphs; our replies follow each respective paragraph. Additionally, the revisions are highlighted in blue in the revised manuscript and marked with 'REV1'.

*The paper is concerned with the estimation of the depth of MIZ affected by the penetration of incoming ocean waves using the altimeter onboard CS2. The wave-affected sea ice regions were identified from two distinctive features of the CS2 waveform characteristics, namely the Stack standard Deviation (SSD) and the Trailing Edge Shape (TES) parameters. An inversion procedure was thus developed and applied in the MIZs of the Atlantic sector of the Arctic over 12 winters from 2010 to 2022. ICESat2 data and Sentinel-1 SAR images were used for comparison to validate the CS2 retrievals.*

*The paper is well-written and the inversion methodology is accurately described. Results are also discussed in comprehensive detail.*

*I have only a few minor remarks as suggestions for the authors:*

*p. 1 last row: Besides frictional processes, wave attenuation in sea ice occurs also as a result of the energy scattering among floes.*

**Reply**: The authors appreciate the reviewer for pointing out the mechanisms of wave attenuation in the marginal ice zone. We have revised the aforementioned statement as follows:

"Furthermore, in the marginal ice zone, wave energy attenuation is predominantly governed by a diversity of processes, which can mainly focus on two mechanisms: dissipation due to interactions between ice floes and the ocean (Doble et al., 2015; Ardhuin et al., 2020; Voermans et al., 2021) and redistribution of energy through the floe-induced wave scattering (Kohout et al., 2006; Squire, 2020)."

*p. 2 rows 35-40: For completeness, it would be useful to mention that spaceborne SAR can image with spatial modes able to distinguish short waves that decay within the first tens of kilometers inside the ice edge of the MIZ. These MIZ regions are typically formed by frazil, grease, and pancake ice, which are becoming the most populated ice types in the Arctic (Wadhams et al. 2018; De Carolis et al. 2021).*

*Wadhams, P., Aulicino, G., Parmiggiani, F., Persson, P. O. G., & Holt, B. (2018). Pancake ice thickness mapping in the Beaufort Sea from wave dispersion observed*

*in SAR imagery. Journal of Geophysical Research: Oceans, 123, 2213–2237. https://doi.org/10.1002/*

*2017JC013003*

*De Carolis, G., Olla, P. & De Santi, F. SAR image wave spectra to retrieve the thickness of grease-pancake sea ice using viscous wave propagation models. Sci Rep 11, 2733 (2021). https://doi.org/10.1038/s41598-021-82228-x*

**Reply:** The author thanks the reviewer for the suggestion of adding SAR-based MIZ observations in this part of the manuscript. We have incorporated the recommended content into the manuscript as follows:

"To resolve waves in the MIZ, high-resolution satellite payloads are typically required, including various optical sensors, Synthetic Aperture Radar (SAR), and laser altimetry of ICESat-2 (IS2) (Markus et al., 2017; Horvat et al., 2020; Collard et al., 2022). These advanced payloads facilitate detailed analysis of sea ice characteristics in the MIZ, including the floe size distribution as well as the wave propagation and attenuation in ice-covered regions(Wadhams et al., 2018; De Carolis et al., 2021; Stopa et al., 2018)".

*p. 7 rows 155-160: How reliable is it to use the sigma0 and its variability information in cases of extreme winds to detect the MIZ boundary?*

**Reply:** The author thanks the reviewer's comment on the feasibility of Sigma_0 for detecting MIZ boundary. We argue that the waveform power of CryoSat-2 (CS2), which characterizes the backscatter at nadir-looking angles (<2-deg), is sufficient to detect the presence of sea ice and distinguish MIZ from open water. Among all the cases we have carried out retrieval, the CS2 waveforms all show drastically higher power on sea ice (MIZ) than the nearby ocean, no matter how strong the waves are on the ocean.

It is worth noting that: the slant-looking backscatter is not suitable for the differentiation between sea ice and open water. Under high ocean conditions, the backscatter on the open ocean is very strong and even higher than that over sea ice. However, over open water at nadir-looking angles, the backscatter mechanism is different from the Bragg-type backscatter at slant-looking angles, which is modulated by wind and the ensuing capillary waves. On the contrary, higher winds (i.e. rougher seas) will slightly reduce the nadir-looking backscatter (instead of increasing it).

Furthermore, the backscatter is very homogeneous over the ocean (along the CS2 track) since the ocean's condition has much larger spatial scales than sea ice. However, a very large variability of backscatter is present over the sea ice-covered

regions due to the backscatter being mostly determined by highly variant snow/surface conditions. To summarize, the CS2 backscatter (along with its distribution) can be used to determine the MIZ's outer boundary.

*p. 9 row 200: "scanning of in the whole..." may be missing a word after "of".*

**Reply**: The author is grateful to the reviewer for identifying the incorrect language in this sentence. It has been revised as follows:
"Second, we scan the entire range of potential value of ξ (from 0 to π, relative to the east)."

*Please revise figure captions: symbols, colored lines, and boxes should be explained in more detail.*

**Reply**: The author appreciates the reviewer's valuable feedback. We will revise the figure captions to include more detailed information of the symbols, colored lines, and boxes for better clarity and understanding. The modifications are highlighted in the revised manuscript.

---

## Author Comment (AC2)

Reply to Review #2:

The authors would like to thank Dr. Boutin for the invaluable comments and suggestions. Below are the replies to each point raised in the review, accompanied by the specific revisions that have been made. The original review comments are presented in *green italic* font and organized in paragraphs; our replies follow each respective paragraph. Additionally, the revisions are highlighted in green in the revised manuscript and marked with 'REV2'.

*This is a review of the manuscript entitled "A 12-Year Climate Record of Wintertime Wave-Affected Marginal Ice Zones in the Atlantic Arctic based on CryoSat-2". The manuscript describes a method to retrieve the wave-affected Marginal Ice Zone (MIZ) using Cryosat- 2. After introducing the importance of the MIZ, they describe their algorithm to retrieve the inner and outer limits of the MIZ. Then, they apply this algorithm over 2 case studies and discuss their definition of the MIZ against wave detected by Sentinel-1 to evaluate their method. They pursue this evaluation using this time a comparison of their method with ICESat-2 retrieved wave-affected MIZ for collocated tracks, with a special focus on 1 case. Having gained confidence in their algorithms and assessed sources of uncertainty, they extend their study to CS2 tracks in the Atlantic Arctic for the period 2010-2022. They describe the properties of the MIZ in 3 sub-regions and find no significant trend in the MIZ width in any of these regions. Finally, they discuss other sources of uncertainty.*

*The manuscript is generally well-written and clear. It synthesizes a large amount of work, with a strong emphasis on the validation of the algorithm using a multiple-sensor approach. The science is sound, well-referenced, and the results are well-discussed. Therefore, I recommend this manuscript for publication after minor revisions.*

**Reply**: The authors thank the reviewer for the comment to our work.

*Minor general comments:*

*I would suggest restructuring section 6 to start with the discussion and end with the summary. I think that would make more sense and conclude the paper on a more "positive" note. I would also suggest concluding by adding a few sentences to give some context to the results. For instance: the dataset is now available to the public and the research community, what type of application do the authors suggest for it? Could we use it already to evaluate the MIZ extent in wave-ice coupled models? What is the next step with this dataset? For instance, is there any plan to retrieve more quantitative data from CS2 in the MIZ (floe size, wave height in ice...)? What is missing to do that? Is there any plan to extend the method to the Pacific Arctic, or Antarctica? Would it work? This conclusion does not need to answer all these*

*questions or to provide an in-depth plan of future work, but I think giving some direction would really improve the impact of the paper.*

**Reply**: Following the reviewer's suggestion, we have reformulated Section 6 to summarize the paper better and introduce the dataset and future work. Specifically, a new section (Sec. 6.4) titled "Summary of the dataset and outlook" is added, which introduces the potential usage of the dataset, potential improvements to it, as well as other aspects of future works on MIZ retrieval.

*I have another general comment that is more like a suggestion. The quantity of information lets me think the manuscript could be divided into two: proof of concept one (section 1 —>4) and a short result article extending section 5. That would certainly increase the impact of section 5 and benefits the authors. Now, the paper is coherent as it is and reads well despite being long, so the decision should be made by the authors.*

**Reply**: The author appreciates the reviewer's thoughtful, constructive suggestions. In particular, we also appreciate the comment on dividing the paper into two distinct parts, with the first containing the proof of concept and the second focusing on the extended results from section 5.

After careful consideration and discussion, we have decided to keep the manuscript as a cohesive unit. Section 5, in its current form, introduces the main retrieval results of the MIZs in the Atlantic Arctic, and it is a key part of the manuscript. Furthermore, based on the dataset, we intend to carry out extended analysis as a future work, which is briefly introduced in Section 6.

*Specific comments:*

*L1: "integral part of the ice cover"—>I am not sure what this expression means. Important part of the ice cover?*

**Reply**: we revise it to: "an essential part of the ice cover".

*L20: I am not sure "incurred" is the right verb here.*

**Reply**: the sentence is revised as: "Consequently, the sea ice cover undergoes complex dynamic and thermodynamic processes, promoting air-sea exchange of heat and moisture within the MIZ."

*L22: Wave attenuation is a big topic and there is no real consensus on which processes (not all related to friction) dominate depending on wave and sea ice conditions. I would suggest "a diversity of processes"? On this note, I may be a bit biased, but I would suggest that a direct application of this dataset is to gain a better understanding of the processes dominating the wave attenuation by constraining the MIZ extent in wave-ice coupled model (see what Boutin et al. 2022 did with Horvat et al. 2020 dataset for instance).*

**Reply**: The author appreciates the reviewer for the comments on the status-quo of our understanding of the wave attenuation. We have revised the aforementioned statement as follows:

"Furthermore, in the marginal ice zone, wave energy attenuation is predominantly governed by a diversity of processes, which can mainly focus on two mechanisms: dissipation due to interactions between ice floes and the ocean (Doble et al., 2015; Ardhuin et al., 2020; Voermans et al., 2021) and redistribution of energy through scattering phenomena caused by sea ice (Kohout et al., 2006; Squire, 2020)."

Besides, we express our hope to apply the new MIZ dataset to both the study of wave attenuation and the validation/intercomparison to the wave-ice coupled model. Although this target is beyond the scope of this paper, we want to mention that the wave attenuation is planned next in our future work. The more detail about the potential application of this new product is now also included in Section 6.

*L23: "more important roles by inducing positive feedback" —> Asplin et al. 2012 only suggest it might be the case, but I don't think it has been proven. I would add potentially (by potentially inducing...).*

**Reply**: The author appreciates the reviewer's suggestion. The sentence is revised as: "MIZs play even more important roles by potentially inducing positive feedback on the sea ice cover".

*L24/25: The sentence is a bit confusing. Also, I'm a bit picky maybe but I feel "Ingvaldsen et al., 2021" is not the best reference to support the statement made here as it discusses physical and ecological changes, not really changes in human activities.*

**Reply**: The authors are grateful to the reviewer for identifying the misleading description of this sentence. Accordingly, the sentence is revised as: "Furthermore, it is also a critical region for human activities, including fishing, tourism, and navigation, due to its distinctive oceanic and ice conditions and unique ecosystem (Palma et al.,2019)." And the new reference is added.

*L32: "and the respective uncertainties" —>the phrasing is confusing here. "and are highly uncertain in the MIZ"? (I am sure there must be a reference for that)*

**Reply**: The author is grateful to the reviewer for identifying the inappropriate language in this sentence. It has been revised as: "... are highly uncertain in the MIZ (Nose et al.,2020)".

*L66: "Furthermore, besides [...] that contain extra information of the ocean's surface." I am a bit confused by that sentence. I would recommend splitting it into shorter simpler sentences.*

**Reply**: The sentence is divided into two shorter ones, as follows: "Furthermore, besides the traditional gated waveform power, the waveform stack describes how the backscatter radar signal for the same footprint changes with different look angles. The waveform stack also contains extra information on the ocean's surface."

*L71: "However, due to the relative coarse resolution of CS2 with respect to the typical wavelengths in MIZs"—>Wavelengths is a bit ambiguous here—>(surface gravity) wave wavelengths.*

**Reply**: We have revised it to: "the wavelength of surface gravity waves" here.

*L74: "Wind waves affect the ice cover by wave/swell generation, the propagation into the ice edge, and the ensuing interaction with sea ice, including breaking the sea ice into smaller floes and the wave attenuation". This sentence is a bit confusing and needs some rephrasing. (For instance, I understand the first part as "Wind waves affect the ice cover because they can generate swells", which is not correct).*

**Reply**: The author is grateful to the reviewer for identifying the inappropriate description of the paragraph. The following paragraph has been revised as follows: "The wind waves and swells, generated from the open ocean, propagate into the ice edge and interact with the sea ice. This process could break the sea ice into smaller floes and further attenuate the wave energy."

*L82 ", waves and swells" —> swells are still waves, so maybe "wind waves and swells"?*

**Reply**: it is revised to "wind waves and swells", which is a more precise description.

*L83: I feel like these references are not the most appropriate to support the statement made here. The fact that waves get longer as they propagate has been known for a while (I'd suggest Robin, 1963, see below).*

**Reply**: We appreciate the reviewer's suggestion for a more proper reference, and we have added the reference to the revised manuscript.

*L87: on—>in ; wave—>waves*

**Reply**: We have revised them accordingly.

*L91: The authors might want to repeat the reference to Figure 1 at the start of this paragraph, it really helps the reader to look at this figure while reading the description of these quantities.*

**Reply**: We have added the reference to Fig. 1 in the sentence: "Therefore, CS2 waveforms on the wave-affected MIZs have the following characteristics (Fig. 1)".

*L107: constitutes*

**Reply**: it is corrected to "constitutes".

*L113: "is utilized" —> I think "is used" works better here, and in a lot of places in the rest of the manuscript.*

**Reply**: The author thanks the reviewer for pointing out the inappropriate language in this sentence. All similar cases in the manuscript have been revised to 'is used'.

*L135: I would recommend referring to a manuscript's figure that shows such patterns (there should be one in Collard et al., 2022 for instance).*

**Reply**: According to the reviewer's suggestion, we have added a reference to Collard et al., (2022) to indicate these patterns.

*L159—>165 I find this paragraph confusing, it could be worth re-ordering the information, maybe starting with the introduction of the physical concept (looking for individual leads as a proxy for pack ice), and then explaining how this is done in*

*practice. I would also recommend adding a comment on this choice of defining pack ice with the presence of leads. Technically, the MIZ can be characterized by the presence of many small leads. While I understand the idea of the authors, I think it can be counter-intuitive to potential readers.*

**Reply**: The author appreciates the reviewer's suggestion on reordering the paragraph. It has been revised as follows:

"Second, among the various waveform parameters, we adopt the SSD as the indicator to determine the along-track transition from the wave-affected part (i.e., the MIZ) to the inner ice pack. To determine the inner boundary of the MIZ, we conducted statistical tests with the distributions of SSD. Specifically, we search for the first lead waveform (available from ESA's baseline product) in the along-track direction and record the sample-based distribution of SSD from the location of the sea ice lead to 100km in length (containing over 300 CS2 footprints). Here, the lead is a flat surface with a high speckle return, observed by CS2. Thus, the wave-affected MIZ cannot extend beyond the location of the first lead. Then, the recorded SSD distribution is used as the benchmark for further determination of the MIZ's inner boundary."

*L200—>203. I find the description of the method to retrieve "xi" hard to follow. I would suggest rewriting it or adding a little schematic.*

**Reply**: We add extra descriptions of the method to retrieve the angle of "xi". The following paragraph has been revised as following:

"Second, we scan the entire range of potential value of $\xi$ (from 0 to $\pi$, relative to the east). For each possible value of $\xi$, we constructed a local intersection line that separated the aforementioned local area into two parts, and computed the accumulated sea ice extent (SIE) for both sides of the intersection line. Then, we defined the final $\xi$ as the angle under which the SIE difference of the two sides is maximum."

*L215: University*

**Reply**: We have corrected it to "University".

*L218: "CS2 measured marked" —> I don't understand.*

**Reply**: In order to make it more clear, we revise it as: "The waveform power measured by CS2 increase ".

*L229: shows—>show*

**Reply**: We have corrected it accordingly.

*L230: "large... than"—>"larger .. than"*

**Reply**: We have corrected it to: "larger .. than".

*L248: "is on the order of"—>"is of the order of"*

**Reply**: We have corrected it accordingly, as: "is of the order of".

*Figure 5: Which green points are associated with panels d,e,f?*

**Reply**: We have revised the figure caption as follows:

"Figure 5. Collocating SAR images from Sentinel-1 (EW mode, panel a) for the MIZ in Fig. 4 and the northern end (red box in panel a) of the CS2-retrieved MIZ shown in detail (panel b). The region with detected wave-in-ice by spectral analysis (Appendix B) on the SAR image is marked by yellow boxes (10km scale). The spectra of the Sentinel-1 backscatter map of three typical regions (green dots in panel a, for the (c)-(e) corresponding to the northernmost, the middle, and the southernmost) are shown on the right, along with the respective fitted parameters and their uncertainties in Eqs. B1. "

*L329: corrected—>correctly?*

**Reply**: We have revised it to: "correctly".

*L354: From the text, I don't understand the reason why the swell penetration is "potentially limited". My guess is that this is because this advected ice is thicker than locally formed one, but this is not clear in the text. Or do the authors mean that there is simply not a large band of ice (and so mechanically a narrow MIZ)? Please clarify.*

**Reply**: We have made revisions of the paragraph to improve its clarity, as follows:

"Sea ice in NS mainly originates from within the Arctic Ocean, due to the ice advection through the transpolar drift and the interaction with the Atlantic inflow. It is usually older and thicker than the locally grown sea ice during the freeze-up season. Consequently, the swell's penetration into the ice pack is potentially limited due to higher ice thickness, and the MIZ is generally narrower in NS. "

*L424: The gridded product resolution is much coarser than the mean width of the MIZ in the Atlantic Arctic. Is it not a problem? I would recommend justifying this choice and detailing what limits the choice of finer resolutions (e.g., the sampling of CS2?).*

**Reply**: The authors would like to make the following clarifications regarding the gridded dataset. First, the choice of the resolution of 2° (zonal) by 1° (meridional) is a trade-off of the CS2 coverage and resolution. For finer resolutions, the CS2 will potentially have insufficient coverage for every gridded location; due to that, the sampling is limited to the nadir locations of the satellite's track. Besides, along the sea ice edge, the representation of MIZ width is usually sufficient at 100 km scale (note the 100km radius for computing SIC-based MIZ width in Fig. 11). We consider the choice of 2° by 1° is proper for characterizing the presence of MIZ.

Second, and more importantly, we consider that the along-track MIZ dataset is the more essential product that we provide. Due to the highly variant nature of MIZs, the monthly or even daily product is insufficient for process-level studies, such as the wave-ice interactions. Such studies require fast and instantaneous sampling of the MIZs, for which only the along-track product is sufficient.

Here, we choose to provide the monthly gridded MIZ dataset together with the along-track product to facilitate potential usages such as climatology analysis and model evaluations.

*References:*

*Robin, G. de Q. (1963). Ocean waves and pack ice. Polar Record, 11(73), 389–393. https://doi.org/10.1017/S003224740005350X*

*Boutin, G., Williams, T., Horvat, C., & Brodeau, L. (2022). Modelling the Arctic wave-affected marginal ice zone: A comparison with ICESat-2 observations. Philosophical Transactions of the Royal Society A: Mathematical, Physical and Engineering Sciences, 380(2235), 20210262. https://doi.org/10.1098/rsta.2021.0262*

**Reply**: These references are added and referred to the revised manuscript accordingly.

---

## Author Response (AR2)

The authors would like to thank the editor and the second round review of our revised manuscript. Following the comments for minor revisions, we have made changes accordingly. This document records all the revisions we have made, listed below. The original comments are in *blue italic*, with our reply following each item of the comments. Moreover, in the marked version of the revised manuscript, the revisions are highlighted with 'REV1'. Additionally, the extra revisions for Language edits are marked out by 'REVLang'.

*Editor Comments:*

*Novel approach to derive the MIZ width of the Atlantic sector in the Arctic from the CryoSat-2 delay-Doppler radar altimeter.*

*Dear authors.*
*Thank you for addressing the reviewers' and editor's comments from the preceding round.*
*Your revised manuscript has been re-reviewed. Pls address the set of comments received from reviewers in response to your revised ms. In additioa, pls also address my comments (below).*

*Editor's Comments:*

*General comments:*
*Figure 1: The top panel does not adequately depict the changes in the wavelength or wave height correctly. Also the ice floes in the MIZ are incorrect, while the sea ice in the Ice Pack is shown as one huge floe on either side of the lead.*

Reply: The author appreciates the reviewer's feedback. We have reviewed and corrected the depiction of the wavelength, wave height, and sea ice representation in Figure 1 to ensure accuracy and clarity.

*l69: Wingham et al. [2006] do not refer to the "sea ice mass balance". Suggest to explore, for example, Ricker et al. [2018; https://doi.org/10.5194/tc-12-3017-2018].*

Reply: The author appreciates the reviewer's suggestion regarding the inappropriate references. We have already added the references here: [Ricker et al., 2018].

*l128L Chnage "Other satellites for the MIZ retrieval" to "Other satellites assisting in the MIZ retrieval"*

Reply: The author appreciates the reviewer's suggestion. The title has been revised to:"Other satellites assisting in the MIZ retrieval"

*l152: Add info as to how long typical Cryo2Ice sampling intervals last.*

Reply: The author appreciates the reviewer's suggestion regarding the addition of information about the typical Cryo2Ice sampling intervals. Accordingly, we have updated the manuscript with the following description:
"Consequently, the ground track of CS2 coincides with that of IS2 at the interval of 19 orbits (about 30 hours), and the average visit interval of the two satellites is within 3 hours(ESA)."

*l185-186: Rewrite "As shown in Figure 1" to describe what is shown in Fig 1 that demonstrates that "larger SSD (...) corresponds to less power drop in the slant looks".*

Reply: The author appreciates the reviewer's suggestion. We have revised the following sentence as follows:
" In the bottom row of Figure 1 and the typical retrieval scenarios in Figure 3 and 4, we show the fitted value of TES from the multilooked waveform and the SSD. SSD is the standard deviation of the Range Integrated Power waveform, with larger values corresponding to slower power decay of the increase in the incidence angle."

*l215: Location names to be capitalized: Change "Greenland sea," to "Greenland Sea,".*

Reply: The author appreciates the reviewer's suggestion. Accordingly, we have revised the location name to "Greenland Sea" and have also reviewed and verified the accuracy of other location names throughout the manuscript.

*l480: The argument "through wave-ice interaction (ice floe breaking, rafting, thermodynamic feedbacks, etc)." holds only if the later MIZ width is larger than the former. Can you show this?*

Reply: The author appreciates the reviewer's suggestion. A brief description of the relevant processes in the MIZ has been added as follows:
"Ice floe breaking, rafting, and thermodynamic feedback collectively accelerate the melting and dynamically expand the MIZ through ice fragmentation and altered ice dynamics (Collins III et al., 2015; Ardhuin et al., 2020)."

*l510: Rewrite "at: https://zenodo.org/record/8176585 (last access: 24 July 2023)" to be completely in brackets. No need for the URL to be in the main sentence.*

Reply: The author appreciates the reviewer's suggestion. We have revised the sentence as follows:

" We provide the MIZ dataset, containing the wintertime MIZs in the Atlantic Arctic region from 2010 to 2022 [https://zenodo.org/record/8176585 (last access: 24 July 2023)]."

*Table 2: Suggest to add information for CS2 retrieval as to how many CS2 tracks were used for each calculation for each region.*

Reply: The author is grateful for the reviewer's suggestion. Accordingly, we have included the number of CS2 tracks utilized for each region in the manuscript.

*Minor comments:*
*\* Correct spelling of "in situ" is without hyphen but in italics. (line 2 and all other uses in this ms)*

Reply: The author appreciates the reviewer's suggestion. We have revised the term "in-situ" to "*in situ*" and formatted it in italics throughout the manuscript.

*l5: Change "ice cover" to "sea ice" or "sea-ice cover".*

Reply: The author appreciates the reviewer's suggestion and has accordingly revised the term "ice cover" to "sea-ice cover".

*l8: Replace "MIZs" with "MIZ events".*

Reply: The author appreciates the reviewer's suggestion and has accordingly revised "MIZs" to "MIZ events".

*l9: Replace "MIZs" with "MIZ events".*

Reply: The author appreciates the reviewer's suggestion. We have revised 'MIZs' to 'MIZ events' and have reviewed the entire manuscript to ensure the accuracy and appropriateness of this description.

*l13-14: Sharpen the statement in "Besides CS2, the proposed retrieval algorithm can be adapted for various historical and future radar altimetry campaigns." -> This is one of the key outcomes of your work described in this ms.*

Reply: The author appreciates the reviewer's suggestion. We have revised the sentence as follows: "Beyond its application to CS2, the proposed retrieval algorithm can be adapted for historical and future radar altimetry campaigns."

*l18: Add "The" to read "The MIZ is...".*

Reply: The author appreciates the reviewer's suggestion and has added "The" to the sentence as recommended.

*l18: "MIZ" has been defined and used in abstract already, suggest to remove here.*

Reply: The author appreciates the reviewer's suggestion and has accordingly removed "marginal ice zone" from the sentence.

*l18: The opening sentence is not informative, as you define MIZ with an undefined concept, the "sea ice edge". I also beg to disagree that the MIZ is deeper than the sea-ice edge.*

Reply: The author appreciates the reviewer's suggestion. We have revised the following sentence as:"The MIZ is on the boundary of the sea-ice covered area that is affected by the open ocean."

*l19: "swells" - Correct to "swell". The noun swell can be countable or uncountable. In more general, commonly used, contexts, the plural form will also be swell.*

Reply: The author appreciates the reviewer's suggestion. We have corrected the "swells" to "swell" in this sentence.

*l25: Add "the" and correct "MIZs" to read "the MIZ plays ... role".*

Reply: The author appreciates the reviewer's suggestion. We have added "the" in this sentence.

*l25: Replace "potentially" with "the likely process" or just cut "potentially".*

Reply: The author appreciates the reviewer's suggestion. We have replaced "potentially" with "the likely process" in the sentence.

*l27: Plural for "ecosystem"??*

Reply: The author appreciates the reviewer's suggestion. We have revised "ecosystem" to "ecosystems" in this sentence.

*l29: Singular for "MIZs", correct to "the MIZ is".*

Reply: The author appreciates the reviewer's suggestion. We have revised "MIZs" to "the MIZ is" in this sentence.

*l30-31: Correct "of the wave's propagation and attenuation in the sea ice cover" to read "of wave propagation into and attenuation by the sea ice".*

Reply: The author appreciates the reviewer's suggestion. We have corrected the sentence to "*In situ* campaigns in MIZs, in spite of the great challenges, provide us with the direct evidence of wave propagation into and attenuation by the sea ice."

*l31: Remove "mainly".*

Reply: The author appreciates the reviewer's suggestion. We have removed "mainly" in this sentence.

*p34-35: Correct "Passive Microwave Imaging (PMI) satellite payloads' to "satellite-borne Passive Microwave Imagers (PMI)".*

Reply: The author appreciates the reviewer's suggestion. We have corrected "Passive Microwave Imaging (PMI) satellite payloads" to "satellite-borne Passive Microwave Imagers (PMI)" in this sentence.

*l36: Correct "ocean's" to "ocean".*

Reply: The author appreciates the reviewer's suggestion. We have corrected "ocean's" to "ocean" in this sentence.

*l37: Correct "fully-packed ice cover" to "compacted sea ice".*

Reply: The author appreciates the reviewer's suggestion. We have corrected "fully-packed ice cover" to "compacted sea ice" in this sentence.

*l38: Correct "close to" to "up to".*

Reply: The author appreciates the reviewer's suggestion. We have corrected "close to" to "up to" in this sentence.

*l41: To be correct add "by satellite-borne instruments" to read "To resolve waves in the MIZ by satellite-borne instruments,"*

Reply: The author appreciates the reviewer's suggestion. We have added "by satellite-borne instruments" in this sentence.

*l41: Rewrite "high-resolution satellite payloads are typically required," to read "atellite payloads providing high spatial resolution are typically required,".*

Reply: The author appreciates the reviewer's suggestion. We have rewritten this sentence to "To resolve waves in the MIZ by satellite-borne instruments, satellite payloads providing high spatial resolution are typically required, including optical sensors, Synthetic Aperture Radar (SAR), and laser altimetry of ICESat2."

*l41: Remove "various".*

Reply: The author appreciates the reviewer's suggestion. We have removed "various".

*l43: Remove "These".*

Reply: The author appreciates the reviewer's suggestion. We have removed "These".

*l45: Rewrite "The effective footprint should be at least finer than half of the wavelength, which is no more than a few hundred meters." to "The spatial resolution of these sensors needs to resolve wavelength in the order of few hundred meters, so in the order of 100 meter."*

Reply: The author appreciates the reviewer's suggestion. We have rewritten the sentence to "The spatial resolution of these sensors needs to resolve wavelength in the order of few hundred meters, so in the order of 100 meter".

*l46: Correct "MIZs" to "MIZ".*

Reply: The author appreciates the reviewer's suggestion. We have corrected "MIZs" to "MIZ".

*l47: Correct "MIZs" to "MIZ".*

Reply: The author appreciates the reviewer's suggestion. We have corrected "MIZs" to "MIZ".

*l47: Change "mainly due to their highly variant nature" to "largely due to its high temporal variability".*

Reply: The author appreciates the reviewer's suggestion. We have changed "mainly due to their highly variant nature" to "largely due to its high temporal variability".

*l48: Correct "MIZs" to "MIZ".*

Reply: The author appreciates the reviewer's suggestion. We have corrected "MIZs" to "MIZ".

*l48: Correct "observation is" to "observations are".*

Reply: The author appreciates the reviewer's suggestion. We have corrected "observation is" to "observations are".

*l49: Remove "potential".*

Reply: The author appreciates the reviewer's suggestion. We have removed "potential".

*l49: Correct "MIZs" to "MIZ".*

Reply: The author appreciates the reviewer's suggestion. We have corrected "MIZs" to "MIZ".

*l50: Correct "MIZs" to "MIZ".*

Reply: The author appreciates the reviewer's suggestion. We have corrected "MIZs" to "MIZ".

*l51: Remove "the region of".*

Reply: The author appreciates the reviewer's suggestion. We have removed "the region of ".

*l52: Replace "including" to "which encompasses the".*

Reply: The author appreciates the reviewer's suggestion. We have corrected "including" to "which encompasses the".

*l52-53: Correct "there exist a variety of sea ice conditions" to "a variety of sea ice conditions exist".*

Reply: The author appreciates the reviewer's suggestion. We have corrected "there exist a variety of sea ice conditions" to "a variety of sea ice conditions exist".

*l53: Remove "the" from "such as the".*

Reply: The author appreciates the reviewer's suggestion. We have removed "the" in this sentence.

*l54: Correct "pass through" to "".*

Reply: The author appreciates the reviewer's suggestion, we have corrected the "pass through" to "develop and enter"

*l55: Correct "Besides.. 2017)." to "Notably, the Atlantic Arctic is rich with human activities, all highly variable due to a numerous dependencies, including those asing from the Atlantification of the region (Polyakov et al., 2017)."*

Reply: The author appreciates the reviewer's suggestion, we have revised the sentence to "Notably, the Atlantic Arctic is rich with human activities, all highly variable due to a numerous dependencies, including those arising from the Atlantification of the region (Polyakov et al., 2017)".

*l57-58" Change "and ... 2022." to read "and derived a 12-winter (2010 - 2022) record for the MIZ in the Atlantic Arctic based on CryoSat-2."*

Reply: The author appreciates the reviewer's suggestion, we have revised the sentence to "and derived a 12-winter (2010 - 2022) record for the MIZ in the Atlantic Arctic based on CryoSat-2".

*l58: Suggest to cut "The paper is organized as follows."*

Reply: The author appreciates the reviewer's suggestion; we have removed "The paper is organized as follows" from the text.

*l60: Replace "typical cases of retrieval" to "case studies".*

Reply: The author appreciates the reviewer's suggestion, we have replaced "typical cases of retrieval" with "case studies" in the manuscript.

*l60: Shorten by removing "Further".*

Reply: The author appreciates the reviewer's suggestion and has removed "Further" from the text.

*l62: Remove "and carries out related analysis".*

Reply: The author appreciates the reviewer's suggestion and has removed "and carries out related analysis".

*l63-64: Remove "Specifically, as shown through intercomparisons, the traditional SIC-based MIZ definition yields much narrower MIZs than our retrieval."*

Reply: The author appreciates the reviewer's suggestion and has accordingly removed the sentence: "Specifically, as shown through intercomparisons, the traditional SIC-based MIZ definition yields much narrower MIZs than our retrieval."

*l64-65: Shorten "Finally, in Section 6 we summarise the paper and discuss related topics of the satellite-based observations of the MIZ." by connecting with the previous sentence.*

Reply: The author appreciates the reviewer's suggestion. We have revised the sentence to read: "Section 5 introduces the 12-year record of the wintertime MIZs in the Atlantic Arctic, while  Section 6 discusses related issues of satellite-based observations of the MIZ. Finally, Section 7 includes a brief summary of the dataset and its potential applications."

*l68: Remove "constantly".*

Reply: The author appreciates the reviewer's suggestion, we have removed "constantly".

*l68: Correct "earth's" to "Earth's".*

Reply: The author appreciates the reviewer's suggestion, we have corrected "earth's" to "Earth's".

*l68: Remove "for over 12 years".*

Reply: The author appreciates the reviewer's suggestion, we have removed  "for over 12 years".

*As discussed with the authors, from here on language issues are not longer listed, instead the authors are referred to a (professional) service for improvements to the text.*

Reply: The author appreciates the reviewer's suggestion. We understand and appreciate the importance of clear and professional language in our manuscript. We had sought the assistance of a professional service to address and improve the language issues as recommended. And the revisions for Language edits are marked out by 'REVLang'.

*l102: Need to specify which "elevation".*

Reply: The author appreciates the reviewer's suggestion, we have revised it as "surface elevation".

*l103-104: Correct style in "Rapley (Rapley, 1984)".*

Reply: The author appreciates the reviewer's suggestion, we have corrected the style to "in Rapley (1984)" in the manuscript.

*l142: Add space " " to read "wavelengths (Collard".*

Reply: The author appreciates the reviewer's suggestion, we have added the space in this sentence.

*l149: Remove "the" to read "Over sea ice,"*

Reply: The author appreciates the reviewer's suggestion, we have removed "the".

*l148: Remove "Compared with the CS2 radar altimeter,".*

Reply: The author appreciates the reviewer's suggestion, we have removed "Compared with the CS2 radar altimeter,".

*l232: Ensure consistent format for dates, i.e., "2015-Feb-17".*

Reply: The author appreciates the reviewer's suggestion, we have checked the format for dates, including in the figure and table, to confirm they are with consistent format.

*l277: Avoid two "back to back" section/subsection headers. Provide a brief statement about using data from other satellites for MIZ validation.*

Reply: The author appreciates the reviewer's suggestion, we have included the following description:

"We validated the MIZ retrieval based on CS2 by conducting a comparative analysis with that derived from the IS2 laser altimeters and the SAR imagery from S1. IS2 and S1 attain high-resolution sampling of the sea ice cover and the MIZ. However, the MIZ retrieval with IS2 is based on its capability to resolve the height signature of waves in the MIZ, whereas that with S1 relies on the wave-modulated backscatter. These methods differ from the proposed CS2-based retrieval methods; hence, they also provide us with complementary perspectives of the processes in the MIZ."

*l375: Remove "We would like to note that,".*

Reply: The author appreciates the reviewer's suggestion. We have removed "We would like to note that,".

*l426: "Summary and discussions": Rename to "Discussion".*

Reply: The author appreciates the reviewer's suggestion. We have renamed "Summary and discussions" to "Discussion".

*l460: Change "More study is needed" to read "Further scientific studies are needed".*

Reply: The author appreciates the reviewer's suggestion. We have corrected *"More study is needed" to "Further scientific studies are needed".*

*l509: Split of the final sub-section "Summary of the dataset and outlook" into "Conclusions".*

Reply: The author appreciates the reviewer's suggestion, we have split the "Summary of the dataset and outlook" into "Conclusions".

*l571: Correct "between 80 m and 800 m" to "between 80 and 800 m".*

Reply: The author appreciates the reviewer's suggestion, we have corrected "between 80 m and 800 m" to "between 80 and 800 m".

*l591: Consider to acknowledge the contribution of the two reviewers.*

Reply: The author appreciates the reviewer's suggestion. We have now included an acknowledgment in our manuscript to recognize the valuable contributions of the editor and two reviewers. Their insights have significantly enriched our work.

---

## Author Response (AR3)

The authors would like to thank the editor for taking the effort of processing the manuscript. Following the comments for technical corrections, we have made revisions accordingly. This document records all the revisions we have made, listed below. The original comments are in *light blue italic*, with our reply following each item of the comments. In the marked version of the revised manuscript, the revisions are highlighted with 'REVEditor'.

*Editor's comments 20240430*

*I acknowledge to significant rewrite of the manuscript and thank the authors for their effort.*

*Minor change:*
*Thoughout ms: Need to be consistent with hyphenation. E.g., "sea-ice cover" (l5) vs "sea ice cover" (l1).*
Reply: The author is grateful to the editor for the suggestion. We have checked the whole manuscript for the consistency of hyphenation.

*l19: Correct "propagate into the ice edge," to "propagate across the ice edge,"*
Reply: The author thanks the editor for identifying the incorrect language in this sentence. It has been revised as "propagate across the ice edge,".

*l20: Correct "Consequently, the sea ice cover undergoes" to "This is a region where the sea ice undergoes".*
Reply: The author thanks the editor for identifying the incorrect language in this sentence. It has been revised as "This is a region where the sea ice undergoes".

*l49: Correct "we use" to "we use data from".*
Reply: The author thanks the editor for identifying the incorrect language in this sentence. It has been revised as "we use data from".

*l90: I am unfamiliar with the concept of a "sea ice lead", instead a lead would be open water or very thin ice, as shown in your Fig. 1. Pls clarify.*
Reply: The author thanks the editor for identifying the inappropriate description in this sentence. We have added the detailed description of lead as: "lead, which would be an area of open water or very thin ice within an expanse of sea ice".

*l148: Correct "we attain 21 collocating track pairs" to "we attain 21 collocated track pairs".*
Reply: The author thanks the editor for identifying the incorrect language in this sentence. It has been revised as "we attain 21 collocated track pairs".

*l148: Reomove "two" from "during the two winters".*

Reply: The author thanks the editor for identifying the incorrect language in this sentence. It has been revised as "during the winters".

*l150: Remove "the" to read "On sea ice".*
Reply: The author thanks the editor for identifying the incorrect language in this sentence. It has been revised as "On sea ice".

*l181: Correct "the multilooked waveform" to "the multilook waveform".*
Reply: The author thanks the editor for identifying the incorrect language in this sentence. It has been revised as "the multilook waveform".

*l226: The date format is unusual in "2015-Feb-17". Pls seek advise from the Editorial Office.*
Reply: The author appreciates the editor's comments. We have revised the date format mentioned in the text of the manuscript to the format "17 February 2015" while maintaining the time format as "2015-Feb-17" on tables and figures to ensure consistency with the formatting of other articles in the journal.

*Fig 3 caption: Correct "The inlet rose map" to "The inset rose map".*
Reply: The author thanks the editor for identifying the incorrect language in this sentence. It has been revised as "The inset rose map".

*Fig 5 (a): Suggest to move the white box with "CS-2" text up (next to the "(a)" label... to avoid covering relevant near-MIZ section of the scene.*
Reply: The author is grateful to the editor for the suggestion. We have changed the location of the white box with "CS-2" text.

*l279: Correct position of the comma in "50 km ,eliminating".*
Reply: The author thanks the editor for identifying the incorrect language in this sentence. It has been revised as "50 km, eliminating".

*l310: Add a space in "indicatingno".*
Reply: The author thanks the editor for identifying the incorrect language in this sentence. It has been revised as "indicating no".

*l320: Correct "21 collocating tracks" to "21 collocated tracks".*
Reply: The author thanks the editor for identifying the incorrect language in this sentence. It has been revised as "21 collocated tracks".

*Fig 6 caption: Correct "collocating tracks" to "collocated tracks".*
Reply: The author thanks the editor for identifying the incorrect language in this sentence. It has been revised as "collocated tracks".

*l331: Add space in "MIZ.CS2".*

Reply: The author thanks the editor for identifying the incorrect language in this sentence. It has been revised as "MIZ. CS2".

*l342: Add space in "Finallyin".*

Reply: The author thanks the editor for identifying the incorrect language in this sentence. It has been revised as "Finally in".

*l347: Remove colon from "are:".*

Reply: The author thanks the editor for identifying the incorrect language in this sentence. The colon has been removed.

*l348: Remove "examples in ".*

Reply: The author thanks the editor for identifying the incorrect language in this sentence. The "examples in " have been removed.

*l352: Correct "years ,such" to "years, such".*

Reply: The author thanks the editor for identifying the incorrect language in this sentence. It has been revised as "years, such".

*l355: Correct "the swell's penetration" to "the swell penetration".*

Reply: The author thanks the editor for identifying the incorrect language in this sentence. It has been revised as "the swell penetration".

*l359: Correct "in GSresulting" to "in the GS resulting".*

Reply: The author thanks the editor for identifying the incorrect language in this sentence. It has been revised as "in the GS resulting".

*l368: Check "nonstormy"... should be "non stormy".*

Reply: The author thanks the editor for identifying the incorrect language in this sentence. It has been revised as "non stormy".

*l427: Change "widest MIZ widths" to "largest MIZ widths".*

Reply: The author thanks the editor for identifying the incorrect language in this sentence. It has been revised as "largest MIZ widths".

*l442: Add space in "advanceand".*

Reply: The author thanks the editor for identifying the incorrect language in this sentence. It has been revised as "advance and".

*l469: Correct "strong forcings" to "strong forcing".*

Reply: The author thanks the editor for identifying the incorrect language in this sentence. It has been revised as "strong forcing".

*Note: I have not rechecked the data links/files. Pls apply honesty approach.*
Reply: *The author appreciates the editor's feedback. We have checked the relevant links and ensured that they can be correctly accessed when mentioned with their latest retrieval time in the article.*

*l557: Rewrite "is slid" in "The local window is slid with a step size of 10 km" for correct English.*
Reply:  The author thanks the editor for identifying the incorrect language in this sentence. It has been revised as "The local window slides with a step size of 10 km"

*l565: Remove "Besides. ".*
Reply:  The author thanks the editor for identifying the incorrect language in this sentence. "Besides. " has been removed here.

*l574-75: Author contributions does NOT explicite show the 4th author.*
Reply: The author thanks the editor for identifying the inappropriate introduction of the author's contributions here, we have added the detailed author contributions of the 4th author.

*l577: For consistency add name "Dr. Petra Heil" to read "the editor Dr. Petra Heil".*
Reply:  The author thanks the editor for identifying the inappropriate description in this sentence. We have revised here with "the editor Dr. Petra Heil".

*l577: Change "and the other anonymous" to "and one anonymous".*
Reply:  The author is grateful to the editor for identifying the inappropriate description in this sentence. We have revised here with "and one anonymous".

*l577: Suggest to move "The authors would like to sincerely thank the editor, as well as Dr. Guillaume Boutin and the other anonymous editor for the invaluable help which significantly improves the paper." to the end of the Acknowledgements.*
Reply:  The author thanks the editor for identifying the inappropriate description in this sentence. This sentence has been moved to the end of the Acknowledgements.

*Tab 1 caption: Corrected "collocating" to "colocated".*
Reply:  The author thanks the editor for identifying the incorrect language in this sentence. It has been revised as "collocated".

*Tab 2 caption: Remove "the period of".*
Reply:  The author thanks the editor for identifying the incorrect language in this sentence. The "the period of" has been removed here.